# Incorporating Visual Correspondence into Diffusion Model for Virtual Try-On

**Siqi Wan** [1]*, **Jingwen Chen**[2], **Yingwei Pan**[2], **Ting Yao**[2], **Tao Mei**[2]
[1] University of Science and Technology of China, [2] HiDream.ai
wansiqi4789@mail.ustc.edu.cn, {chenjingwen, pandy, tiyao, tmei}@hidream.ai

## Abstract

Diffusion models have shown preliminary success in virtual try-on (VTON) task. The typical dual-branch architecture comprises two UNets for implicit garment deformation and synthesized image generation respectively, and has emerged as the recipe for VTON task. Nevertheless, the problem remains challenging to preserve the shape and every detail of the given garment due to the intrinsic stochasticity of diffusion model. To alleviate this issue, we novelly propose to explicitly capitalize on visual correspondence as the prior to tame diffusion process instead of simply feeding the whole garment into UNet as the appearance reference. Specifically, we interpret the fine-grained appearance and texture details as a set of structured semantic points, and match the semantic points rooted in garment to the ones over target person through local flow warping. Such 2D points are then augmented into 3D-aware cues with depth/normal map of target person. The correspondence mimics the way of putting clothing on human body and the 3D-aware cues act as semantic point matching to supervise diffusion model training. A point-focused diffusion loss is further devised to fully take the advantage of semantic point matching. Extensive experiments demonstrate strong garment detail preservation of our approach, evidenced by state-of-the-art VTON performances on both VITON-HD and DressCode datasets. Code is publicly available at: https://github.com/HiDream-ai/SPM-Diff.

## 1 Introduction

Virtual Try-ON (VTON) is an increasingly appealing direction in computer vision field, that aims to virtually drape the provided garment items onto target human models. The task empowers the end users to experience the visual affects of wearing various clothings without the need of physical store try-ons. That has a great potential impact for revolutionizing the shopping experience within E-commerce industry. VTON can be regarded as one kind of conditional image synthesis Song et al. (2023); Yu et al. (2023); Chen & Kae (2019); Cong et al. (2020); Li et al. (2023a); Chen et al. (2024a;b; 2023a) with two constrains (i.e., the given in-shop garment and the target person image). However, the typical conditional image synthesis commonly tackles spatially aligned conditions like human pose or sketch/edges Zhang et al. (2023b); Mou et al. (2024). In contrast, VTON signifies a flexible change in shape of in-shop garment in real world, thereby being more challenging due to the complexities associated with the preservation of intrinsic clothing geometry and appearance (i.e., global garment shape and local texture details).

When generative networks become immensely popular, the early VTON techniques Choi et al. (2021); Lee et al. (2022); Xie et al. (2023); Chopra et al. (2021); Dong et al. (2019b); Ge et al. (2021) consider capitalizing on explicit warping to deform in-shop garment according to the pose of the target person, leading to spatially-aligned condition of warped garment. These aligned conditions are further fed into Generative Adversarial Networks (GAN) Goodfellow et al. (2014); Karras et al. (2021; 2019; 2020b;a) for person image generation. Nevertheless, it is found that such GAN-based methods (e.g., GP-VTON Xie et al. (2023)) show unrealistic artifacts especially when the garment texture is complex and the human pose is challenging as in Figure 1 (b). This can be attributed to the warping errors of complex textures in garment and the limited generative capacity of GAN for synthesizing real-world

---

*This work was performed at HiDream.ai.

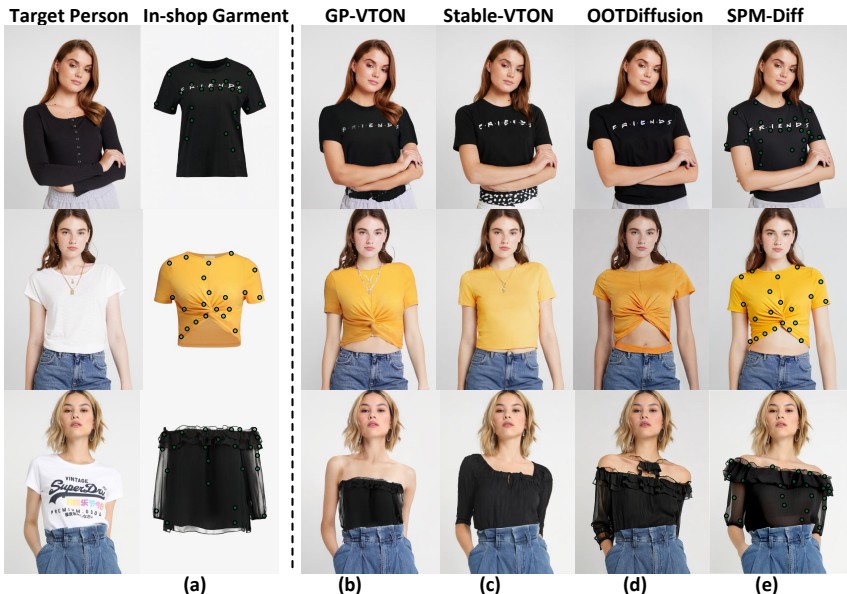

Figure 1: Illustration of given target person and (a) in-shape garment with semantic points. Existing GAN-based methods (e.g., (b) GP-VTON) and diffusion-based approaches (e.g., (c) Stable-VTON and (d) OOTDiffusion) often struggle with complex garment texture details and challenging human poses, resulting in a range of artifacts and the lack of necessary texture details. In contrast, (e) our SPM-Diff effectively alleviates these limitations and leads to higher-quality results with better-aligned semantic points, leading to strong visual correspondence and thereby preserving garment detail/shape.

person images. To tackle this issue, recent advances Gou et al. (2023); Morelli et al. (2023); Xu et al. (2024a); Kim et al. (2024) take the inspiration from Diffusion models Avrahami et al. (2022); Hertz et al. (2023); Kawar et al. (2023); Ruiz et al. (2023); Rombach et al. (2022) with enhanced training stability and scalability in content creation, and present a new diffusion-based direction for VTON task. In general, the whole garment image as appearance reference is encoded via VAE encoder and UNet Kingma & Welling (2014); Ronneberger et al. (2015), which is further integrated into diffusion model for conditional image generation. Despite improving the quality of synthesized person image with few artifacts, such VTON results still fail to preserve sufficient garment details (Figure 1 (c-d)) due to the stochastic denoising process in diffusion model.

Unlike previous efforts that rely on the whole garment to trigger diffusion process, we view VTON problem from a new perspective of visual correspondence in the paradigm of diffusion model. Intuitively, each in-shop garment image contains interest points, i.e., 2D locations in an image which are stable and repeatable from different viewpoints. In analogy to the traditional interest points in geometric computer vision field Sun et al. (2021); Hedlin et al. (2023); Tang et al. (2023); Wang et al. (2023), we name such kind of interest point in VTON task as "semantic point." As shown in Figure 1 (a), when viewed individually, each semantic point refers to the unique regional fine-grained texture detail; when viewed as a whole, all semantic points reflect the holistic garment shape. Both regional fine-grained texture detail and holistic garment shape derived from semantic points are supposed to be nicely preserved for VTON tasks. In other words, these semantic points should be aligned with the visually corresponding ones in the synthesized person image. That motivates us to introduce the explicit correspondences of semantic points between in-shop garment and output synthetic person image to diffusion model. With this semantic point matching, we can temper the stochasticity of diffusion model, leading to higher-quality VTON results with better-aligned garment shape and texture details (Figure 1 (e)).

By consolidating the idea of capitalizing on visual correspondence prior for virtual try-on, we present a novel diffusion model with Semantic Point Matching (SPM-Diff). Specifically, SPM-Diff first samples a set of semantic points rooted in the given garment image, and matches them to points on target human body according to garment-to-person correspondence via local flow warping. Furthermore, these 2D appearance cues of semantic points are augmented into 3D-aware cues with depth/normal map of target person, mimicking the way of putting clothing on human body. Such 3D-aware cues are later injected into a dual-branch diffusion framework to facilitate virtual try-on. In an effort to amplify the semantic point matching along the whole diffusion process, we exquisitely

devise a point-focused diffusion loss that puts more focus on the reconstruction of semantic points over target persons. Empirical results on two VTON benchmarks demonstrate the evident superiority of our SPM-Diff on garment detail and shape preservation against the state-of-the-art methods.

## 2 RELATED WORK

**GAN-based Virtual Try-on.** To tackle virtual try-on (VTON) task, prior works Wang et al. (2018); Li et al. (2021); Fele et al. (2022); Morelli et al. (2022); Dong et al. (2019b); He et al. (2022); Choi et al. (2021); Lee et al. (2022); Xie et al. (2023) typically adopt a two-stage strategy, first deforming the garment to fit the target body shape and then integrating the transformed garment onto the human model using a GAN-based image generator to synthesize the final person image capitalizing on conditions like the warped garment and human pose. One key point for these works is to accurately warp the garment to the clothing region. As one of the pioneer works, VITON Han et al. (2018) estimates thin-plate spline transformation (TPS) Bookstein (1989) based on hand-craft shape context to achieve realistic garment deformation. Later, CP-VTON Wang et al. (2018) introduces an upgraded learnable TPS transformation to boost the performances. However, the results remain far from satisfactory due to artifacts in the misaligned areas between the suboptimally deformed garment and the desired clothing region, particularly as image resolution increases for online shopping. To tackle this problem, VITON-HD Choi et al. (2021) predicts a segmentation map of the desired clothing regions to guide VTON synthesis, which is utilized to alleviate the impact from the misaligned regions. Recently, HR-VTON Lee et al. (2022) novelly introduces a try-on condition generator that combines garment warping and segmentation modules to resolve visual misalignment and occlusion. Moreover, an innovative Local-Flow Global-Parsing warping module is proposed in GP-VTON Xie et al. (2023), which warps garments parts individually and assembles locally warped results based on the global garment parsing. Despite the promising results, these methods fail to handle complex poses and maintain fine-grained garment details due to the limited generative capability of GANs.

**Diffusion-based Virtual Try-on.** Diffusion models Ho et al. (2020); Rombach et al. (2022); Chen et al. (2023b); Zhang et al. (2024); Zhu et al. (2024); Qian et al. (2024); Yang et al. (2024) have drawn widespread attention due to their superior capability in visual generation compared to GANs Goodfellow et al. (2014); Pan et al. (2017). Therefore, some works Gou et al. (2023); Morelli et al. (2023); Zhu et al. (2023); Kim et al. (2024); Xu et al. (2024a) have attempted to incorporate diffusion models into the pipeline for VTON tasks, striving to generate a photo-realistic image that preserves appearance patterns. For example, DCI-VTON Gou et al. (2023) directly overlays the deformed garment over the target person with the desired clothing regions masked out to steer the diffusion models. Furthermore, LaDI-VTON Morelli et al. (2023) additionally learns token embeddings of the given garment and refines the garment detials with an upgraded decoder. GarDiff Wan et al. (2024) simultaneously employs CLIP Radford et al. (2021) and Variational Auto-encoder (VAE) Kingma & Welling (2014) to encode the reference garment into appearance priors, which are regarded as additional conditions to guide the diffusion process. However, solely capitalizing on the deformed garment could introduce notable artifacts due to the suboptimal warping results. Therefore, Stable-VTON Kim et al. (2024) novelly improves controlnet Zhang et al. (2023b) with zero cross-attention blocks to implicitly deform the input garment and further injects the intermediate features extracted by the proposed module into the UNet of diffusion models to boost VTON. Most recently, a dual-branch framework with two UNets is presented Xu et al. (2024a); Choi et al. (2024) to fully leverage the pre-trained image prior in the diffusion models for garment feature learning. In this framework, a reference-UNet is initialized from the pre-trained one (dubbed Main-UNet) and employed to learn multi-scale features of the input garment, while the Main-UNet refers to these features for high-fidelity image generation. Although appealing results are achieved by these methods, it remains challenging to preserve every detail of the garment due to weak visual correspondence between the garment and synthesized person using implicit warping mechanism.

## 3 METHOD

In this section, we first briefly review the fundamental concepts of Latent Diffusion Model in Sec. 3.1. Next, we elaborate technical details of the overall framework of our proposed SPM-Diff and the novel semantic point matching (SPM) in Sec. 3.2.1 and 3.2.2, respectively. Finally, an upgraded training objective to facilitate semantic point matching is demonstrated in Sec. 3.3.

### 3.1 PRELIMINARY

The prevalent Latent Diffusion Model (LDM) Rombach et al. (2022) is adopted as our foundational model in this work. Specifically, LDM first exploits a pre-trained Variational Autoencoder Kingma & Welling (2014) $\mathcal{E}(\cdot)$ to map an input image $I_0$ from the high-dimensional pixel space to the latent space: $\mathbf{x}_0 = \mathcal{E}(I_0)$. In the forward diffusion process at timestep $t \sim \mathcal{U}(0, T)$, noises are added to the latent code $\mathbf{x}_0$ according to a pre-defined variance schedule $\{\beta_s\}_1^T$ as follows:

$$\mathbf{x}_t = \sqrt{\bar{\alpha}_t}\mathbf{x}_0 + \sqrt{1 - \bar{\alpha}_t}\epsilon, \qquad (1)$$

where $\epsilon \sim \mathcal{N}(0, 1)$, $\bar{\alpha}_t = \prod_{s=1}^t (1 - \beta_s)$, and $\beta_t \in \{\beta_s\}_1^T$. In the reverse denoising process, LDM learns to predict the added noise $\epsilon$ and removes it. The typical training objective of LDM (generally implemented as a UNet Ronneberger et al. (2015)) parameterized by $\theta$ can be simply formulated as

$$\mathcal{L}(\theta, \mathbf{x}_0, \mathbf{c}) = \mathbb{E}_{\epsilon, t}[||\epsilon_\theta(\mathbf{x}_t, \mathbf{c}, t) - \epsilon||_2^2], \qquad (2)$$

where $\mathbf{c}$ denotes the conditioning text prompt. During inference, LDM starts from a random noisy latent code and iteratively denoises it given the text prompt step by step for image generation.

### 3.2 OUR SPM-DIFF

In virtual try-on (VTON) task, given a target person image $I_p \in \mathbb{R}^{H \times W \times 3}$ and an in-shop garment $I_g \in \mathbb{R}^{H' \times W' \times 3}$, the model is required to synthesize a high-quality image $I \in \mathbb{R}^{H \times W \times 3}$, where the person $I_p$ wears the in-shop garment $I_g$. The main challenge lies in preserving the intricate details and shape of the in-shop garment, due to the high diversity of synthetic contents (e.g., varied human body) within the intricate sampling space of diffusion models.

#### 3.2.1 OVERALL FRAMEWORK

Recently, the utilization of Reference-Net has demonstrated remarkable efficacy in retaining the fine-grained details of reference images across diverse fields Cao et al. (2023); Hu et al. (2024); Nam et al. (2024); Xu et al. (2024b), shedding new light on VTON. Drawing inspirations from these studies, we frame our SPM-Diff within a basic dual-branch architecture comprising two UNets: Main-UNet and Garm-UNet, where the Main-UNet refers to the features of the implicitly deformed garment from the Garm-UNet for image generation. However, suboptimal outcomes are observed in our experiments with this trivial dual-branch network. We argue that the reason is Main-UNet fails to establish precise visual correspondence between the given garment in $I_g$ and the synthesized person in the output image $I$ by simply using the appearance features from Garm-UNet. Consequently, the synthesized garment is prone to inconsistent fine-grained details with the input garment. Therefore, we propose to explicitly incorporate visual correspondence prior into the diffusion model to boost VTON. Specifically, a novel semantic point matching (SPM) is additionally introduced to our SPM-Diff. In SPM, we interpret the fine-grained appearance and texture details as a set of local semantic points, which denote the locations on the garment that are stable and repeatable from different view points. These semantic points are matched to the corresponding ones on the target person through local flow warping. These 2D cues are then converted into 3D-aware cues by depth/normal map augmentation, which act as visual correspondence prior to enhance Main-UNet for detail preservation.

Specifically, the semantic point set $P_G$ on the given garment is first sampled and mapped into the corresponding one $P_H$ on the target person via local flow warping. Then, the multi-scale point features of $P_G$ / $P_H$ are extracted by a pre-trained garment/geometry feature encoder, which are further augmented and injected into the Main-UNet through the proposed SPM module. Formally, given the latent code $\mathbf{x}_t$, the denoised $\mathbf{x}_{t-1}$ predicted by our SPM-Diff can be computed as

$$\mathbf{x}_{t-1} = \frac{1}{\sqrt{\alpha_t}}(\mathbf{x}_t - \frac{1 - \alpha_t}{\sqrt{1 - \bar{\alpha}_t}}\hat{\epsilon}) + \sigma_t\epsilon,$$

$$\hat{\epsilon} = \epsilon_\theta(\mathbf{x}_t, \mathbf{x}_a, \mathbf{x}_d, \mathbf{x}_n, \mathbf{x}_{I_g}, \mathbf{c}_{I_g}, P_G, P_H, t), \qquad (3)$$

where $\alpha_t = 1 - \beta_t$, $\sigma_t^2 = \beta_t$. $\theta$ are the parameters of our SPM-Diff. $\mathbf{x}_n$ and $\mathbf{x}_d$ are the latent codes of normal map $I_n$ and depth map $I_d$ of the target person, respectively. $\mathbf{x}_{I_g}$ and $\mathbf{c}_{I_g}$ denote the latent code and CLIP image embedding of the in-shop garment $I_g$, respectively. $\mathbf{x}_a$ is the latent code of a garment-agnostic image $I_a$ that is derived by applying a mask over the potential regions in $I_p$ to be filled with the input garment. Fig. 2(b) illustrates the overview of our SPM-Diff.

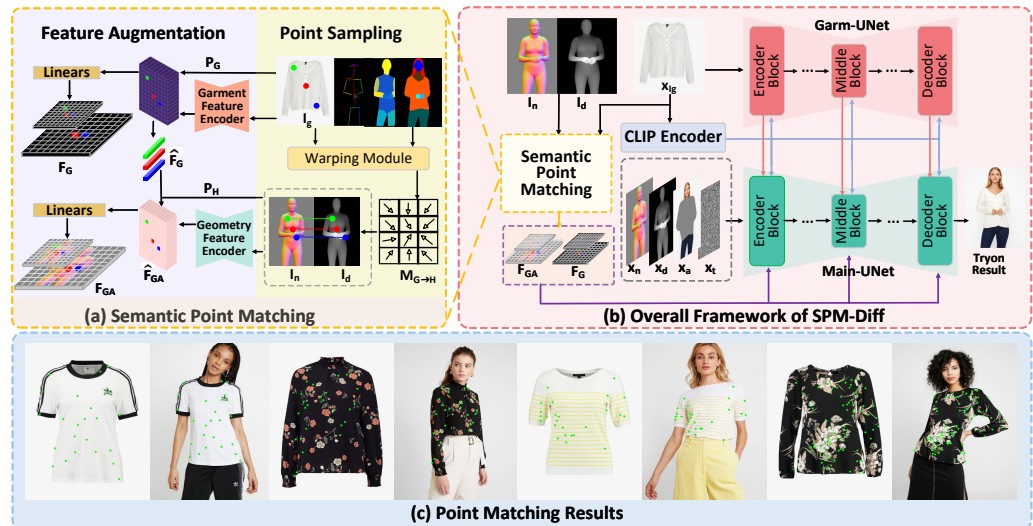

Figure 2: The overall framework of our SPM-Diff. (a) Illustration of our semantic point matching (SPM). In SPM, a set of semantic points on the garment are first sampled and matched to the corresponding points on the target person via local flow warping. Then, these 2D cues are augmented into 3D-aware cues with depth/normal map, which act as semantic point matching to supervise diffusion model. (b) Dual-branch framework includes Garm-UNet and Main-UNet for garment feature learning and image generation, respectively. Note that Main-UNet is upgraded with SPM for high-fidelity synthesis in our SPM-Diff. (c) Visualization of garment-to-person point correspondence.

### 3.2.2 SEMANTIC POINT MATCHING

As shown in Fig. 2(a), a set of local semantic points $P_G$ (e.g., red, green, blue points) on the in-shop garment are first sampled and projected to the corresponding points $P_H$ on the depth/normal map $I_d$ / $I_n$ of the target person via local flow warping. Note that the garment-agnostic depth/normal map is derived by rendering from an estimated SMPL model of the target person Lin et al. (2023). The descriptive point features $\hat{\mathbf{F}}_G$ / $\hat{\mathbf{F}}_H$ of the points in $P_G$ / $P_H$ are extracted by a pre-trained garment/geometry feature encoder. Then, the 2D representations $\hat{\mathbf{F}}_G$ are further augmented into 3D-aware ones $\hat{\mathbf{F}}_{GA}$ that perceive the body shape of the target person by fusing with $\hat{\mathbf{F}}_H$. Finally, the derived features $\hat{\mathbf{F}}_G$ and $\hat{\mathbf{F}}_{GA}$ are integrated into the Main-UNet to boost VTON.

**Point Sampling.** In order to faithfully retain the local fine-grained details (e.g., texture, shape) of the in-shop garment, sparse semantic points of interest are sampled for affordable computation overhead. Specifically, Superpoint DeTone et al. (2018) is employed to sample a set of $N$ pixel-level interest points $P^N$ on the garment image $I_g$. Then, the farthest point sampling strategy is adopted to select a subset of $K$ semantic points from $P^N$, denoted as $P_G = \{(x^k, y^k) \in \mathbb{R}^2 | x^k \in [1, H'], y^k \in [1, W'], k = 1...K\}$, that approximates $P^N$ with fewer points while preserving the important characteristics of the garment. Note that $(x^k, y^k)$ is the 2D coordinate on the image $I_g$.

Then, we associate the semantic points $P_G$ on the garment $I_g$ with the corresponding ones $P_H$ on the target person $I_p$ through a dense displacement field $\mathbf{M}_{G \to H} \in \mathbb{R}^{H \times W \times 2}$. Each element $\mathbf{M}_{G \to H}^{(i,j)}$ represents the relative displacement (i.e., 2D offset vector) for each point at coordinate $(i, j)$ on in-shop garment $I_g$ relative to the person image $I_p$. Here we employ flow warping module like GP-VTON Xie et al. (2023) to estimate the dense displacement field in between. More details of our flow warping module will be depicted in the Appendix A.4. Then we can retrieve the corresponding semantic points $P_H$ on the target person $I_p$ by applying $\mathbf{M}_{G \to H}$ to the point set $P_G$ as follows:

$$P_H(x^k, y^k) = P_G(x^k, y^k) + \mathbf{M}_{G \to H}^{(x^k, y^k)}. \tag{4}$$

We visualize some 2D garment-to-person semantic point pairs from $(P_G, P_H)$ in Fig. 2 (c).

**Feature Augmentation.** The features $\hat{\mathbf{F}}_G$ / $\hat{\mathbf{F}}_H$ of the point sets $P_G$ / $P_H$ are first extracted from a pre-trained garment/geometry feature encoder. The garment feature encoder and geometry feature encoder share the similar architecture of Garm-UNet and Main-UNet, respectively. Next, $\hat{\mathbf{F}}_G$ is

further augmented by the $\hat{\mathbf{F}}_H$ into $\hat{\mathbf{F}}_{GA}$:

$$\hat{\mathbf{F}}_{GA}(x_k, y_k) = \hat{\mathbf{F}}_G(x_k, y_k) + \hat{\mathbf{F}}_H(x_k, y_k). \tag{5}$$

As such, the 2D cues of the semantic points are transformed into 3D-aware ones, mimicking the way garment changes shape with human body and enabling better visual correspondence between the garment and the target person.

**Feature Injection.** To effectively facilitate semantic point matching for high-fidelity VTON, we inject the derived features $\hat{\mathbf{F}}_{GA}$ and $\hat{\mathbf{F}}_G$ into the self-attenion layers of Main-UNet in a multi-scale manner. Specifically, a series of MLPs $\phi_{GA} = \{\phi_{GA}^l\}_{l=1}^L$ and $\phi_G = \{\phi_G^l\}_{l=1}^L$ are exploited to project $\hat{\mathbf{F}}_{GA}$ and $\hat{\mathbf{F}}_G$, respectively, into the multi-scale features $\mathbf{F}_{GA}$ and $\mathbf{F}_G$ that match the feature dimensions of $L$ self-attention layers in the Main-UNet as follows:

$$\mathbf{F}_{GA}^l(x_k, y_k) = \phi_{GA}^l(\hat{\mathbf{F}}_{GA}(x_k, y_k)), \quad \mathbf{F}_G^l(x_k, y_k) = \phi_G^l(\hat{\mathbf{F}}_G(x_k, y_k)). \tag{6}$$

Given the geometry features $\mathbf{F}_{GA}^l$ and garment features $\mathbf{F}_G^l$ of semantic points, we first augment intermediate hidden states of Main-UNet/Garm-UNet (i.e., $\mathbf{K}^l/\mathbf{K}_g^l$) with $\mathbf{F}_{GA}^l/\mathbf{F}_G^l$. Then the augmented intermediate features of Main-UNet/Garm-UNet are concatenated, followed with $l$-th self-attention layer in Main-UNet (Please refer to Fig. 16 in Appendix for more details of the feature injection process):

$$Attn(\mathbf{Q}^l, \mathbf{K}_*^l, \mathbf{V}^l, \mathbf{F}_{GA}^l, \mathbf{F}_G^l) = Softmax(\frac{(\mathbf{Q}^l + \mathbf{F}_{GA}^l) \cdot [\mathbf{K}^l + \mathbf{F}_{GA}^l, \mathbf{K}_g^l + \mathbf{F}_G^l]^T}{d} \mathbf{V}^l),$$
$$\mathbf{Q}^l = W_m^{Q,l} \mathbf{h}_t^l, \quad \mathbf{K}^l = W_m^{K,l} \mathbf{h}_t^l, \quad \mathbf{K}_g^l = W_g^{K,l} \mathbf{h}_g^l, \quad \mathbf{V}^l = [W_m^{V,l} \mathbf{h}_t^l, \; W_g^{V,l} \mathbf{h}_g^l], \tag{7}$$

where $\mathbf{h}_t^l$ and $\mathbf{h}_g^l$ are the intermediate hidden states from the Main-UNet and Garm-UNet, respectively. $W_m^{Q,l}$, $W_m^{K,l}$ and $W_m^{V,l}$ are the projection matrices in the $l$-th self-attention layer of the Main-UNet. $W_g^{K,l}$ and $W_g^{V,l}$ are the projection matrices in the $l$-th self-attention layer of the Garm-UNet. $[\cdot]$ is the concatenation operation over the token sequence. In practice, the point features $\mathbf{F}_G^l / \mathbf{F}_{GA}^l$ are repositioned onto an empty spatial feature map according to the interpolated coordinates of $P_G / P_H$ for efficient feature addition as shown in leftmost part of Fig. 2 (a). Since the middle layers in Main-UNet perceive high-level visual concepts instead of the fine-grained garment details, only the first two and last two self-attention layers are considered in point feature injection.

By explicitly incorporating the visual correspondence between the garment and the target person and further augmenting the 2D cues with 3D geometric conditions via the proposed semantic point matching, our SPM-Diff can better preserve the local garment details and shape more accurately.

### 3.3 TRAINING OBJECTIVE

To amplify the semantic point matching along the whole diffusion process, we devise a point-focused diffusion loss that emphasizes on the reconstruction of semantic points over target persons by increasing the loss weight of each semantic point. Thus, the final training objective is

$$\mathcal{L}_{point\_focused} = ||\epsilon - \hat{\epsilon}||_2^2 + \lambda ||(\epsilon - \hat{\epsilon}) \odot M(P_H)||_2^2, \tag{8}$$

where $\hat{\epsilon}$ is defined as in Eq. 3, and $M(P_H)$ denotes a binary mask that only triggers on the semantic points $P_H$ and $\lambda$ is the trade-off coefficient.

## 4 EXPERIMENTS

### 4.1 EXPERIMENTAL SETUPS

**Datasets.** We train our model on two virtual try-on benchmarks, VITON-HD Choi et al. (2021) and DressCode Morelli et al. (2022). VITON-HD dataset contains 13,679 frontal-view woman and upper garment image pairs. Following the general practices of previous works Gou et al. (2023); Morelli et al. (2023), the dataset is divided into two disjoint subsets: a training set with 11,647 pairs and a testing set with 2,032 pairs. DressCode dataset consists of 53,795 image pairs, which are categorized into three macro-categories: 15,366 for upper-body clothes, 8,951 pairs lower-body clothes, and

Table 1: Quantitative results in single dataset evaluation on VITON-HD and DressCode upper-body (D.C.Upper). The subscript $*_{sdxl}$ indicates the use of superior base model (stable diffusion xl).

| Train/Test | VITON-HD/VITON-HD | | | | D.C.Upper/D.C.Upper | | | |
|---|---|---|---|---|---|---|---|---|
| Method | SSIM ↑ | LPIPS ↓ | FID ↓ | KID ↓ | SSIM ↑ | LPIPS ↓ | FID ↓ | KID ↓ |
| **PF-AFN** Ge et al. (2021) | 0.888 | 0.087 | 9.654 | 1.04 | 0.910 | 0.049 | 17.653 | 5.43 |
| **HR-VTON** Lee et al. (2022) | 0.878 | 0.105 | 12.265 | 2.73 | 0.936 | 0.065 | 13.820 | 2.71 |
| **SDAFN** Bai et al. (2022) | 0.880 | 0.082 | 9.782 | 1.11 | 0.907 | 0.053 | 12.894 | 1.09 |
| **FS-VTON** He et al. (2022) | 0.886 | 0.074 | 9.908 | 1.10 | 0.911 | 0.050 | 16.470 | 4.22 |
| **GP-VTON** Xie et al. (2023) | 0.884 | 0.081 | 9.072 | 0.88 | 0.769 | 0.268 | 20.110 | 8.17 |
| **LaDI-VTON** Morelli et al. (2023) | 0.864 | 0.096 | 9.480 | 1.99 | 0.915 | 0.063 | 14.262 | 3.33 |
| **Stable-VTON** Kim et al. (2024) | 0.852 | 0.084 | 8.698 | 0.88 | 0.911 | 0.050 | 11.266 | 0.72 |
| **DCI-VTON** Gou et al. (2023) | 0.880 | 0.080 | 8.754 | 1.10 | **0.937** | 0.042 | 11.920 | 1.89 |
| **OOTDiffusion** Xu et al. (2024a) | 0.881 | 0.071 | 8.721 | 0.82 | 0.906 | 0.053 | 11.030 | 0.29 |
| **IDM-VTON** Choi et al. (2024) | 0.877 | 0.082 | 9.079 | 0.79 | 0.891 | 0.065 | 10.860 | 0.32 |
| **IDM-VTON**$_{sdxl}$ Choi et al. (2024) | 0.916 | 0.061 | 7.033 | 0.53 | 0.926 | 0.040 | **9.561** | 0.16 |
| **SPM-Diff** | 0.911 | 0.063 | 8.202 | 0.67 | 0.927 | 0.042 | 10.560 | 0.19 |
| **SPM-Diff**$_{sdxl}$ | **0.917** | **0.055** | **6.871** | **0.52** | 0.933 | **0.038** | 9.622 | **0.14** |

29,478 for dresses. As in the original splits, 1,800 image pairs from each category are used for testing, and the remaining pairs are used as training data.

Additionally, we leverage a human image dataset (SSHQ-1.0 Fu et al. (2022)) to evaluate our method beyond standard virtual try-on datasets. Note that we follow Stable-VTON Kim et al. (2024) and adopt the first 2,032 images for evaluation in SHHQ-1.0.

**Evaluation.** We evaluate the performances in two testing settings, i.e., paired setting and unpaired setting. The paired setting uses a pair of a person and the original clothes for reconstruction, whereas the unpaired setting involves changing the clothing of a person image with different clothing item. Meanwhile, we adopt single dataset evaluation that performs training and evaluation within a single dataset, and cross-dataset evaluation implies extending our evaluation over different datasets. The experiments on all settings are conducted at the resolution of $512 \times 384$.

In the paired setting, the input garment is highly correlated to the one depicted in primary person image. Hence we directly follow the standard evaluation setup to leverage Structural Similarity (SSIM) Wang et al. (2004) and Learned Perceptual Image Patch Similarity (LPIPS) Simonyan & Zisserman (2015) for measuring the visual similarity between the generated image and the ground-truth one. In the unpaired setting, since the given target person originally wears a different garment from the input in-shop garment and the ground-truth try-on results are unavailable, only the Fréchet Inception Distance (FID) Heusel et al. (2017) and Kernel Inception Distance (KID) Bińkowski et al. (2018) can be used to evaluate the quality of outputs.

**Implementation Details.** In our SPM-Diff, Garm-UNet and Main-UNet are initialized from the pre-trained Stable Diffusion 1.5, which are further finetuned over VTON datasets. The garment/geometry feature encoder shares similar model structure as Garment/Main-UNet. We jointly pre-train these two encoders in a basic dual-branch network with additional depth/normal map as inputs but without our proposed SPM mechanism, i.e., the ablated run Base+Geo map in Table 4. Then, the pre-trained garment/geometry feature encoder is utlized to extract garment/geometry point features, which are further injected into the self-attention layers of Main-UNet in our SPM-Diff. We employ AdamW Loshchilov & Hutter (2019) ($\beta_1 = 0.9$, $\beta_2 = 0.999$) to optimize the model. The learning rate is set to 0.00005 with linear warmup of 500 iterations. The hyper-parameter $\lambda$ in Equation (8) is set to 0.5. OpenCLIP ViT-H/14 Ilharco et al. (2021) is utilized to extract the CLIP visual embeddings of the input garment. To enable classifier-free guidance Ho & Salimans (2022), the embeddings of the garment and depth/normal map are randomly dropped with a probability of 0.1. We train SPM-Diff on a single Nvidia A100 GPU with 45,000 iterations (batch size: 16). At inference, the output person image is progressively generated via 20 steps with a UniPC Zhao et al. (2023) sampler, and the scale of classifier-free guidance is set as 2.0. Compared with a trivial dual-branch model, the computation overhead is slightly increased with SPM-Diff: 1.97 secs (SPM-Diff) vs 1.50 secs (OOTDiffusion).

## 4.2 BENCHMARK RESULTS

**Quantitative Results in Single Dataset Evaluation.** We first present the VTON results on VITON-HD and DressCode (three macro-categories) in Table 1 and 2 under in-distribution setup, where both training and testing data are derived from the same dataset/category. Note that for fair comparison with IDM-VTON$_{sdxl}$ that adopts superior base model (Stable Diffusion xl), we include additional

Table 2: Quantitative results in single dataset evaluation on DressCode lower-body (D.C.Lower) and DressCode dresses (D.C.Dresses) datasets. The subscript $*_{sdxl}$ indicates the use of superior base model (stable diffusion xl).

| Train/Test | D.C.Lower/D.C.Lower | | | | D.C.Dress/D.C.Dress | | | |
|---|---|---|---|---|---|---|---|---|
| Method | SSIM ↑ | LPIPS ↓ | FID ↓ | KID ↓ | SSIM ↑ | LPIPS ↓ | FID ↓ | KID ↓ |
| **PF-AFN** Ge et al. (2021) | 0.903 | 0.056 | 19.683 | 6.89 | 0.875 | 0.074 | 19.257 | 7.66 |
| **HR-VTON** Lee et al. (2022) | 0.912 | 0.045 | 16.345 | 4.02 | 0.865 | 0.113 | 18.724 | 4.98 |
| **SDAFN** Bai et al. (2022) | 0.913 | 0.049 | 16.008 | 2.97 | 0.879 | 0.082 | 12.362 | 1.28 |
| **FS-VTON** He et al. (2022) | 0.909 | 0.054 | 22.031 | 7.46 | 0.888 | 0.073 | 20.821 | 8.02 |
| **GP-VTON** Xie et al. (2023) | 0.923 | 0.042 | 16.65 | 2.86 | 0.886 | 0.072 | 12.65 | 1.84 |
| **LaDI-VTON** Morelli et al. (2023) | 0.911 | 0.051 | 13.38 | 1.98 | 0.868 | 0.089 | 13.12 | 1.85 |
| **OOTDiffusion** Xu et al. (2024a) | 0.892 | 0.049 | 9.72 | 0.64 | 0.879 | 0.075 | 10.65 | 0.54 |
| **DCI-VTON** Gou et al. (2023) | 0.924 | 0.035 | 12.34 | 0.91 | 0.887 | 0.068 | 12.25 | 1.08 |
| **IDM-VTON** Choi et al. (2024) | 0.919 | 0.041 | 12.05 | 0.93 | 0.881 | 0.074 | 12.33 | 1.41 |
| **IDM-VTON**$_{sdxl}$ Choi et al. (2024) | **0.934** | 0.032 | 8.77 | 0.44 | 0.904 | **0.060** | 9.87 | 0.47 |
| **SPM-Diff** | 0.932 | 0.034 | 9.02 | 0.48 | 0.899 | 0.063 | 10.17 | 0.50 |
| **SPM-Diff**$_{sdxl}$ | **0.934** | **0.030** | **8.68** | **0.42** | **0.909** | **0.060** | **9.62** | **0.44** |

Table 3: Quantitative results in cross dataset evaluation. We train VTON models on one of VITON-HD and DressCode upper-body (D.C.Upper), and then evaluate them on different datasets. The subscript $*_{sdxl}$ indicates the use of superior base model (stable diffusion xl).

| Train/Test | VITON-HD/D.C.Upper | | | | D.C.Upper/VITON-HD | | | | VITON-HD/SSHQ-1.0 | |
|---|---|---|---|---|---|---|---|---|---|---|
| Method | SSIM ↑ | LPIPS ↓ | FID ↓ | KID ↓ | SSIM ↑ | LPIPS ↓ | FID ↓ | KID ↓ | FID ↓ | KID ↓ |
| **HR-VTON** Lee et al. (2022) | 0.862 | 0.108 | 24.170 | 7.35 | 0.811 | 0.228 | 45.923 | 36.69 | 52.732 | 31.22 |
| **GP-VTON** Xie et al. (2023) | 0.897 | 0.385 | 12.451 | 6.67 | 0.804 | 0.262 | 22.431 | 27.90 | 20.99 | 8.76 |
| **LaDI-VTON** Morelli et al. (2023) | 0.901 | 0.101 | 16.336 | 5.36 | 0.801 | 0.243 | 31.790 | 23.02 | 24.904 | 6.07 |
| **DCI-VTON** Gou et al. (2023) | 0.905 | 0.142 | 17.3426 | 12.03 | 0.825 | 0.179 | 16.670 | 6.40 | 24.850 | 6.68 |
| **Stable-VTON** Kim et al. (2024) | 0.914 | 0.065 | 13.182 | 2.26 | 0.855 | 0.118 | 10.104 | 1.70 | 21.077 | 5.10 |
| **OOTDiffusion** Xu et al. (2024a) | 0.915 | 0.061 | 11.965 | 1.20 | 0.839 | 0.123 | 10.220 | 2.72 | 19.62 | 5.45 |
| **IDM-VTON** Choi et al. (2024) | 0.909 | 0.066 | 12.398 | 1.18 | 0.821 | 0.145 | 10.091 | 2.88 | 18.78 | 4.88 |
| **IDM-VTON**$_{sdxl}$ Choi et al. (2024) | 0.925 | 0.052 | 10.003 | 0.89 | 0.879 | 0.086 | 8.203 | 1.98 | 14.32 | 2.67 |
| **SPM-Diff** | 0.922 | 0.055 | 10.057 | 1.01 | 0.844 | 0.101 | 9.990 | 2.70 | 16.78 | 3.99 |
| **SPM-Diff**$_{sdxl}$ | **0.929** | **0.050** | **9.235** | **0.65** | **0.882** | **0.059** | **8.196** | **1.62** | **13.68** | **2.60** |

variant of our SPM-Diff (i.e., SPM-Diff$_{sdxl}$) that uses the same base model of Stable Diffusion xl. Overall, for VITON-HD and each category in DressCode, our SPM-Diff consistently achieves better results against existing VTON approaches across most metrics, including both GAN-based models (HR-VTON, SDAFN, GP-VTON, etc) and diffusion-based models (e.g.,LaDI-VTON, Stable-VTON, OOTDiffusion, etc). In particular, on VITON-HD dataset, the SSIM score in paired setting and the FID score in unparied setting of our SPM-Diff can reach 0.911 and 8.202, which leads to the absolute gain of 0.030 and 0.519 against the strong competitor OOTDiffusion. The results basically demonstrate the key merit of taming diffusion model with semantic point matching mechanism to facilitate virtual try-on task. More specifically, diffusion-based methods commonly exhibit better unpaired scores than GAN-based models. This observation is not surprising, since diffusion model reflects more powerful generative capacity than GAN in image synthesis. In between, instead of simply encoding in-shop garment via CLIP as additional condition in DCI-VTON, Stable-VTON and OOTDiffusion capitalize on VAE encoder and UNet to achieve more fine-grained appearance representations for triggering diffusion process, thereby leading to better VTON results. Nevertheless, the performances of them are still inferior to our SPM-Diff, since the visual correspondence between in-shop garment and output synthetic person image is under-explored in existing diffusion-based approaches. Instead, by novelly formulating VTON task as semantic pointing matching problem, SPM-Diff tempers the stochasticity of diffusion model with explicit visual correspondence prior, and achieves competative VTON results.

**Quantitative Results in Cross Dataset Evaluation.** Proceeding further, we evaluate all VTON models under out-of-distribution setup in Table 3, where training and testing data are derived from different datasets. Similar to the observations in single dataset evaluation, our SPM-Diff still exhibits better performances than other baselines across most metrics. The results again demonstrate the advantage of modeling semantic point matching for virtual try-on.

**Qualitative Results.** Next, we perform a qualitative evaluation of different methods through case study. Fig. 3 shows some examples of VITON-HD dataset. Overall, our SPM-Diff can better preserve fine-grained garment details and the contour of the synthesized garment across varied human body shapes compared with other baselines. This verifies the merit of incorporating semantic matching prior and garment feature augmentation with 3D geometric conditions into diffusion model. For example, both GAN-based models (GP-VTON) and diffusion-based models (e.g., Stable-VTON,

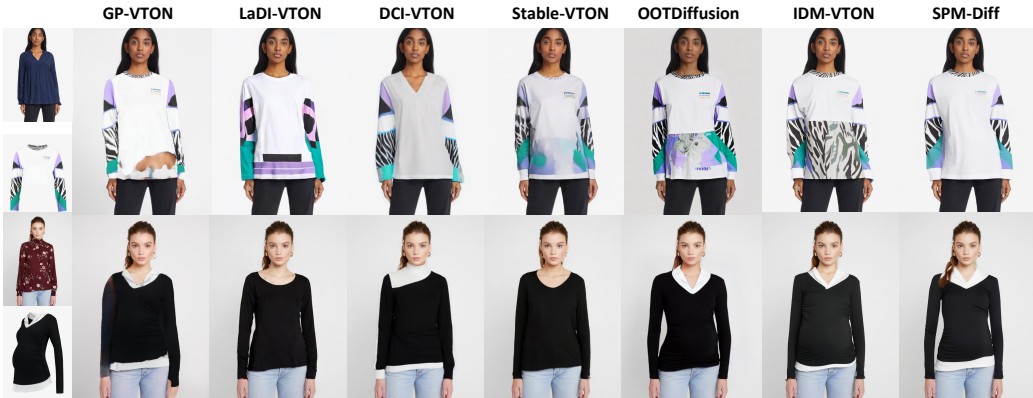

Figure 3: Qualitative results on the VITON-HD dataset.



Figure 4: User study on 100 garment-person pairs randomly sampled from VITON-HD dataset. OOTDiffusion, etc) fail to retain accurate garment details on 1st row, while our SPM-Diff successfully restores the appearance of the garment. Please refer to Appendix A.1 for more qualitative results.

**User Study.** A comprehensive user study is conducted to examine whether the synthesized images conform to human preferences. We randomly sample 100 unpaired garment-person from VITON-HD dataset for evaluation. 10 participants from diverse education backgrounds (i.e., fashion design (4), journalism (2), psychology (2), business (2)) are invited and asked to choose the winner between our SPM-Diff and the competing approaches based on three criteria: 1) preservation of the fine-grained appearance details, 2) preservation of the garment contour across diverse human body shapes. 3) the perceived image realism. Fig. 4 summarizes the averaged results by all the participants of different methods over 100 garment-person pairs. As shown in Fig. 4, our SPM-Diff significantly outperforms the rival models by a large margin regarding garment detail, shape preservation and image realism.

## 4.3 DISCUSSIONS

**Effect of Pivotal Components in SPM-Diff.** Here, we investigate the effect of each pivotal component in our SPM-Diff. We consider one more design at each run and the overall performances on VITON-HD dataset are listed in Table. 4. **Base** is implemented as the trivial dual-branch framework with two UNets described in Sec. 3.2.1, which takes $(\mathbf{x}_t, \mathbf{x}_a, \mathbf{x}_{I_g}, \mathbf{c}_{I_g})$ as input. By incorporating the depth and normal maps (denoted as **Geo map** in Table. 4) as additional conditions, **Base + Geo map** better controls the body pose and shape of the synthesized person, yielding marginal improvements over **Base**. **Base + Geo map + SPM** significantly boosts the performances by enforcing the alignment of semantic points between the garment and target person and further augmenting the 2D features of these points with 3D geometric conditions. Particularly, **Base + Geo map + SPM** makes the relative gains of 16.9% and 27.7% on LPIPS and KID against **Base + Geo map**, respectively. This again demonstrates the effectiveness of capturing better visual correspondence for high-fidelity VTON synthesis. Moreover, the point-focused diffusion loss (**P-loss**) is devised to emphasize the reconstruction of semantic points over target persons, and best results are attained by **Base + Geo map + SPM + P-loss**. Some examples generated by each run are illustrated in the Appendix A.2.

**Effect of Semantic Point Count in SPM.** We then analyze the effect of semantic point count $\mathbf{K}$. Table. 5 summarizes the performances of our SPM-Diff with varying $\mathbf{K} \in [10, 25, 50, 75]$ on VITON-HD dataset. When fewer semantic points ($\mathbf{K} = 10$) are sampled, the fine-grained garment details may be not fully captured, which hinders the preservation of all the garment details in the synthesized images. Increasing $\mathbf{K}$ from 10 to 25 alleviates this aforementioned issue and improves the results. However, employing more semantic points ($\mathbf{K} = 50$ or 75) yields worse results. Specifically, more

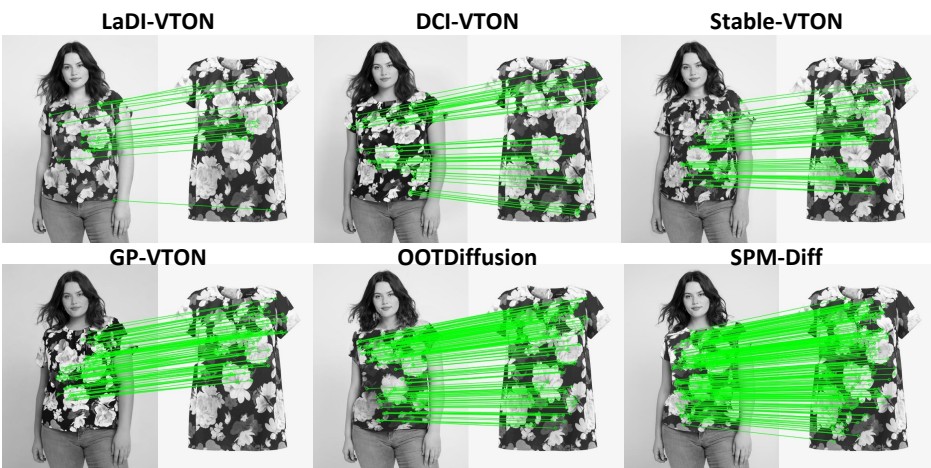

Figure 5: Accuracy of visual correspondence between in-shop garment and synthesized person image.

Table 4: Ablation study of our proposed SPM-Diff on the VITON-HD dataset.

| Model | SSIM ↑ | LPIPS ↓ | FID ↓ | KID ↓ |
|---|---|---|---|---|
| (1) **Base** | 0.876 | 0.088 | 8.830 | 1.01 |
| (2) **Base + Geo map** | 0.881 | 0.083 | 8.621 | 0.94 |
| (3) **Base + Geo map + SPM** | 0.904 | 0.069 | 8.320 | 0.68 |
| (4) **Base + Geo map + SPM + P-loss** | **0.911** | **0.063** | **8.202** | **0.67** |

Table 5: Ablation study of semantic point count **K** on the VITON-HD dataset.

| Point Count K | SSIM ↑ | LPIPS ↓ | FID ↓ | KID ↓ |
|---|---|---|---|---|
| **K = 10** | 0.890 | 0.083 | 8.689 | 0.85 |
| **K = 25** | **0.911** | **0.063** | **8.202** | **0.67** |
| **K = 50** | 0.901 | 0.070 | 8.503 | 0.73 |
| **K = 75** | 0.893 | 0.081 | 8.599 | 0.78 |

semantic points are inevitably clustered in local neighborhoods in pixel space with large **K**, which will be interpolated to same locations in the low-dimensional feature map. However, their corresponding points projected onto the target person may be different due to the significant offsets from imprecise local flow map $\mathbf{M}_{G \to H}$, leading to one-to-many point misalignment and compromising the holistic coherence of the synthesized images. Some examples are visualized in the Appendix A.2.

**Effect of SPM in Visual Correspondence.** In addition, we evaluate the visual correspondence accuracy between in-shop garment and synthesized person image, which can be generally regarded as one kind of detail-preserving capability. We follow the typical point mapping method in Super-Point DeTone et al. (2018) to perform HPatches homography estimation. Specifically, we perform nearest neighbor matching between all interest points and their descriptors detected in the in-shop garment image and those in the synthesized person image. Then we employ OpenCV's implementation of findHomography() with RANSAC to compute the correctly matched points in the image pairs, which are further marked in green (Fig. 5). In this way, the denser the green point mappings covered over two images, the better the perservation of interest points (i.e., garment details) for synthesized person image. It can be easily observed that our SPM-Diff manages to align more semantic points and therefore preserve more fine-grained garment details than the other competing baselines, which validates our proposal of semantic point matching.

## 5 CONCLUSION

We have presented SPM-Diff, a new diffusion-based model for virtual try-on task. Different from holistically encoding in-shop garment as appearance reference, SPM-Diff uniquely excavates visual correspondence between the garment and synthesized person through semantic points of interest rooted in the given garment. Such structured prior is fed into diffusion model to facilitate garment detail preservation along diffusion process. Extensive experiments validate the superiority of our SPM-Diff when compared to state-of-the-art VTON approaches in terms of both single dataset evaluation and cross dataset evaluation.

**Acknowledgement.** This work was supported in part by the Beijing Municipal Science and Technology Project No. Z241100001324002 and Beijing Nova Program No. 20240484681.

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

## A   APPENDIX

### A.1   MORE QUALITATIVE RESULTS

Fig. 6 showcases more examples generated by different methods on the VITON-HD dataset, while the visual results of other clothing categories (i.e., lower-body clothes and dresses) on the DressCode dataset are summarized in Fig. 7.

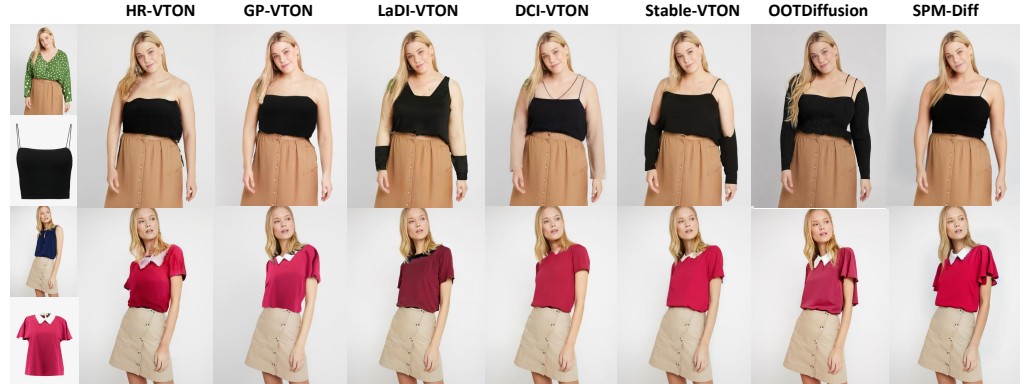

Figure 6: More qualitative results on the VITON-HD dataset.

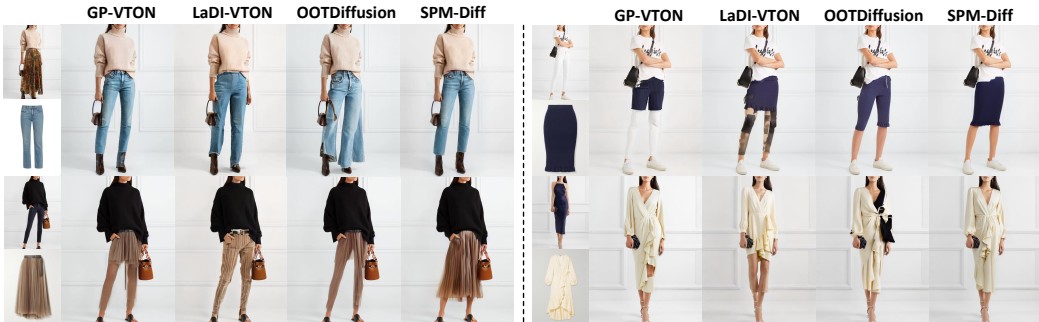

Figure 7: Qualitative results on the DressCode dataset.

### A.2   DISCUSSIONS

**Effect of Pivotal Components in SPM-Diff.** Fig. 8 illustrates some examples generated by the different runs in Table. 4.

**Effect of Semantic Point Count in SPM.** Fig. 9 takes an example to demonstrate the effect of semantic point count $K$. Similar to the observation in Table 5, the most visually pleasing result is achieved when $K = 25$. We agree that significantly increasing semantic point count (e.g., from $K = 25$ to $K = 50$ or 75) will result in redundancy and interpolation/projection error, leading to degraded results as discussed in the main paper. However, when taking an in-depth look at the sensitivity of semantic point count around the optimal value $K = 25$, the semantic point count is not sensitive. To validate this, we experimented by varying semantic point count $K$ in the range of $[20, 40]$ within an interval of 5 on VITON-HD dataset. As shown in Table 7, the performance under each metric only fluctuates within the range of 0.1, which basically validates that the performance is not sensitive to the change of semantic point count within this range.

**Effect of Hyper-parameter $\lambda$ in Eq.(8).** Table 6 summarizes the results of our SPM-Diff with different hyper-parameter $\lambda$ on VITON-HD dataset, and $\lambda = 0.5$ achieves the best performances across all the metrics.

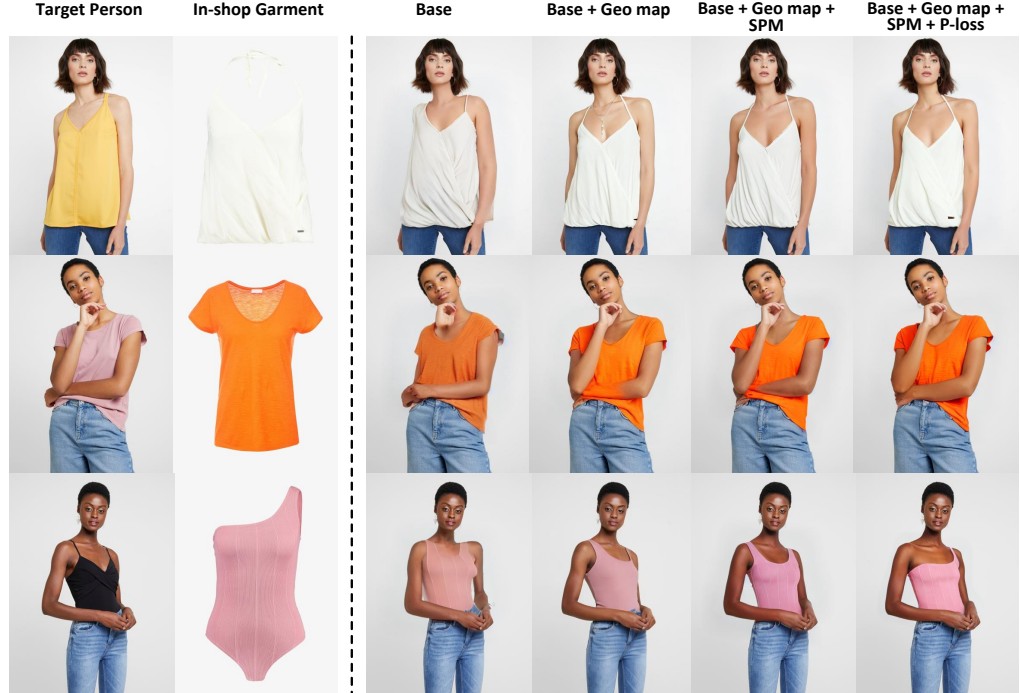

Figure 8: Ablation study of the pivotal components in SPM-Diff on VITON-HD dataset.

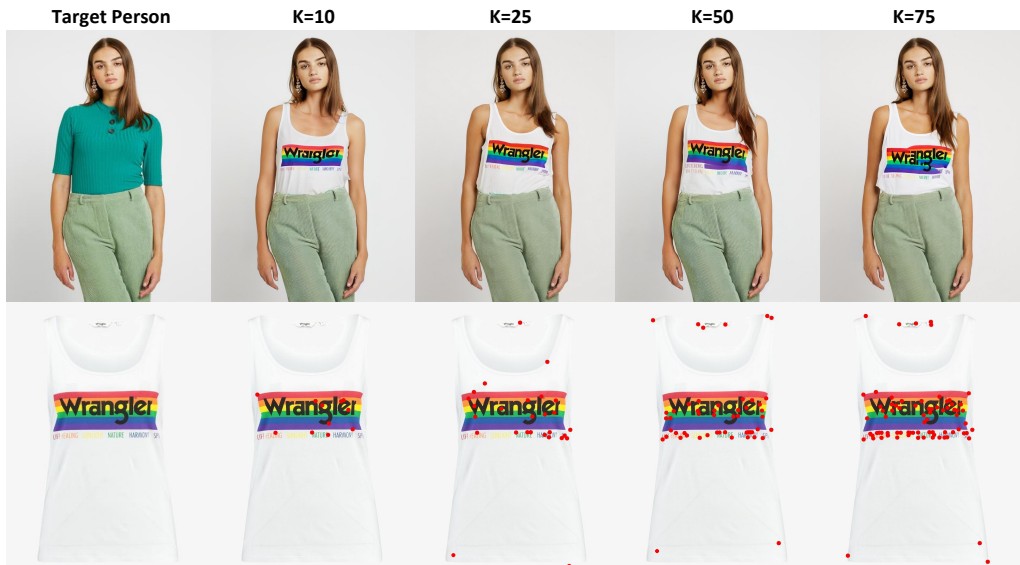

Figure 9: Ablation study of semantic point count **K** on VITON-HD dataset.

**Evaluation of 3D Conditions Learnt by Various Methods.** The 3D cues (i.e., depth/normal map) adopted in our training and inference are the estimated results by using pre-trained 3D human reconstruction model (OSX Lin et al. (2023)), rather than manually annotated accurate 3D conditions. We conducted experiments by evaluating SPM-Diff with varied estimated 3D cues achieved from different 3D human reconstruction models (e.g., PyMAF-X Zhang et al. (2023a), OSX Lin et al. (2023), 4DHumans Goel et al. (2023)) on VITON-HD dataset. The results are summarized in Table 8, and similar performances are attained across different pre-trained 3D human reconstruction models.

**Generalization to In-the-Wild VTON.** Moreover, to further evaluate the generalization of our SPM-Diff under challenging scenario, we take in-the-wild person images with complex backgrounds as inputs (in contrast to the clean backgrounds in VITON-HD dataset). As shown in Fig. 10, our SPM-Diff manages to achieve promising results.

Table 6: Effect of hyper-parameter $\lambda$ in Eq.(8).

| $\lambda$ | SSIM ↑ | LPIPS ↓ | FID ↓ | KID ↓ |
|---|---|---|---|---|
| 1.0 | 0.899 | 0.071 | 8.318 | 0.689 |
| 0.5 | **0.911** | **0.063** | **8.202** | **0.670** |
| 0.1 | 0.910 | 0.066 | 8.287 | 0.687 |
| 0.05 | 0.906 | 0.067 | 9.309 | 0.682 |
| 0.01 | 0.905 | 0.067 | 8.320 | 0.680 |

Table 7: Ablation study of semantic point count K on the VITON-HD dataset.

| Point Count K | SSIM ↑ | LPIPS ↓ | FID ↓ | KID ↓ |
|---|---|---|---|---|
| **K = 20** | 0.903 | 0.061 | 8.201 | 0.66 |
| **K = 25** | 0.911 | 0.063 | 8.202 | 0.67 |
| **K = 30** | 0.908 | 0.060 | 8.210 | 0.65 |
| **K = 35** | 0.905 | 0.062 | 8.228 | 0.63 |
| **K = 40** | 0.889 | 0.066 | 8.215 | 0.64 |

Table 8: Quantitative results with varied estimated 3D cues achieved from different 3D human reconstruction models on VITON-HD dataset.

| 3D human reconstruction model | SSIM ↑ | LPIPS ↓ | FID ↓ | KID ↓ |
|---|---|---|---|---|
| **PyMAF-X Zhang et al. (2023a)** | 0.901 | 0.064 | 8.265 | 0.63 |
| **OSX Lin et al. (2023)** | 0.911 | 0.063 | 8.202 | 0.67 |
| **4DHumans Goel et al. (2023)** | 0.908 | 0.060 | 8.211 | 0.68 |

**Discussion with Warp-based Methods.** In particular, existing warp-based methods Dong et al. (2019a); Han et al. (2019); Xie et al. (2023); Li et al. (2023b) commonly adopt warping model to warp the input garment according to the input person image, and then directly leverage the warped garment image as hard/strong condition to generate VTON results. This way heavily relies on the accuracy of garment warping, thereby easily resulting in unsatisfactory VTON results given the inaccurate warped garments under challenging human poses (see the results of GP-VTON in Fig. 11). On the contrary, our proposed SPM-Diff adopts a soft way to exploit the visual correspondence prior learnt via the warping model as a soft condition to boost VTON. Technically, our SPM-Diff injects the local semantic point features of the input garment into the corresponding positions on the human body according to a flow map estimated by the warping model. This way nicely sidesteps the inputs of holistic warped garment image with amplified pixel-level projection errors, and only emphasizes the visual correspondence of the most informative semantic points between in-shop garment and output synthetic person image. Note that such visual correspondence of local semantic points are exploited in latent space (corresponding to each local region), instead of precise pixel-level location. Thus when encountering mismatched points with slight displacements within local region, SPM-Diff still leads to similar geometry/garment features as matched points, making the visual correspondence more noise-resistant (i.e., more robust to warping results). As shown in Fig. 11, given warping results with severe distortion, our SPM-Diff still manages to achieve promising results, which basically validate the effectiveness of our proposal.

**Evaluation of Point-matching Capability.** we examine the point-matching capability of the pre-trained warping module adopted in our SPM-Diff, in comparison with a vanilla stable diffusion model. Note that here we use a training-free approach (DIFT Tang et al. (2023)) as baseline that exploits the stable diffusion model to match the corresponding points between two images. Specifically, we construct the point-matching test set by manually annotating semantic points on a subset of garment images and their corresponding points on the person image. Next, given the target semantic points on the garment image, our adopted warping module and DIFT perform point matching on the person image. Finally, we report the mean square error (MSE) between the matched points via each run and the annotated ground truth. As shown in the Table 9, our adopted warping module demonstrates stronger point-matching capability than the vanilla stable diffusion model. In addition, we visualize the point-matching results of each run in Fig. 12, and our adopted warping module yields more accurate point-matching results than vanilla stable diffusion model.

Table 9: Evaluation of the point-matching method.

| Model | DIFT Tang et al. (2023) | Our Warping Model |
|---|---|---|
| $MSE$ (↓) | 39.46 | 29.84 |

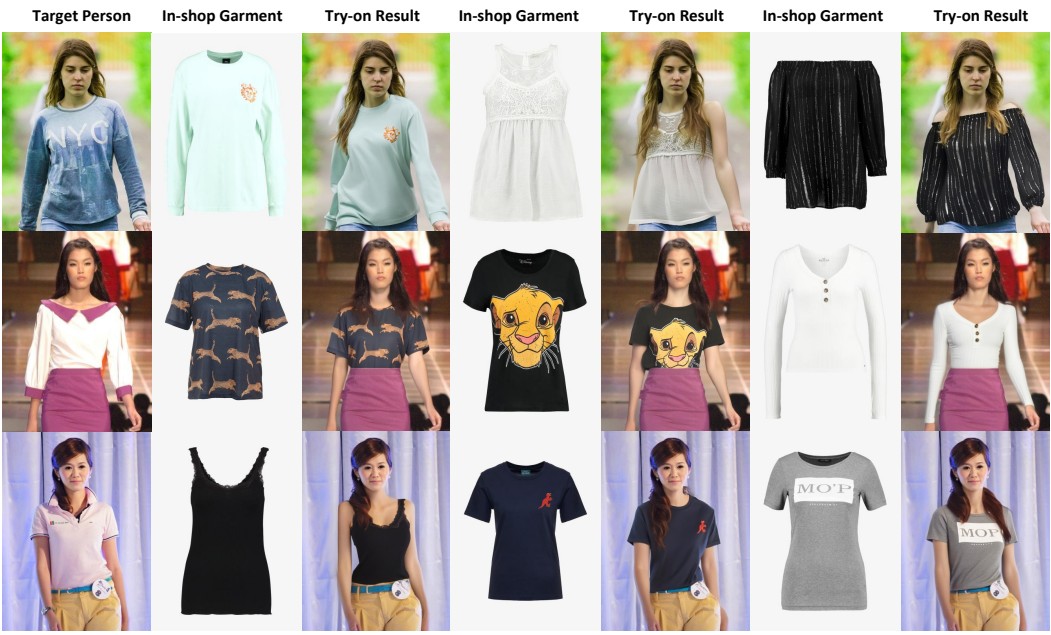

Figure 10: Tryon results on in-the-wild person images with complex backgrounds.

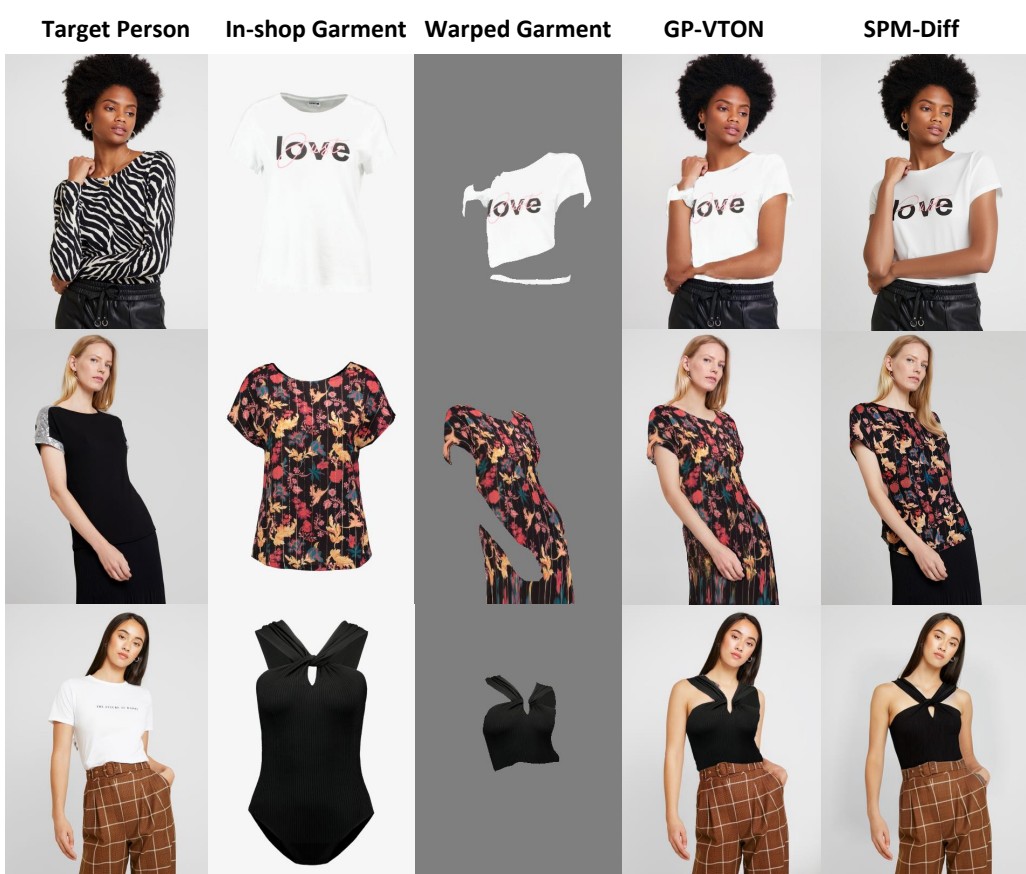

Figure 11: Comparisons with warp-based methods on the VITON-HD dataset.

**Discussion on Semantic Point Matching for Loose Garments.** For the cases of loose garments like dresses or skirts, some semantic points of the garment may fail to exactly fall on the human body. However, these points still convey certain local details of the garment (garment features), and

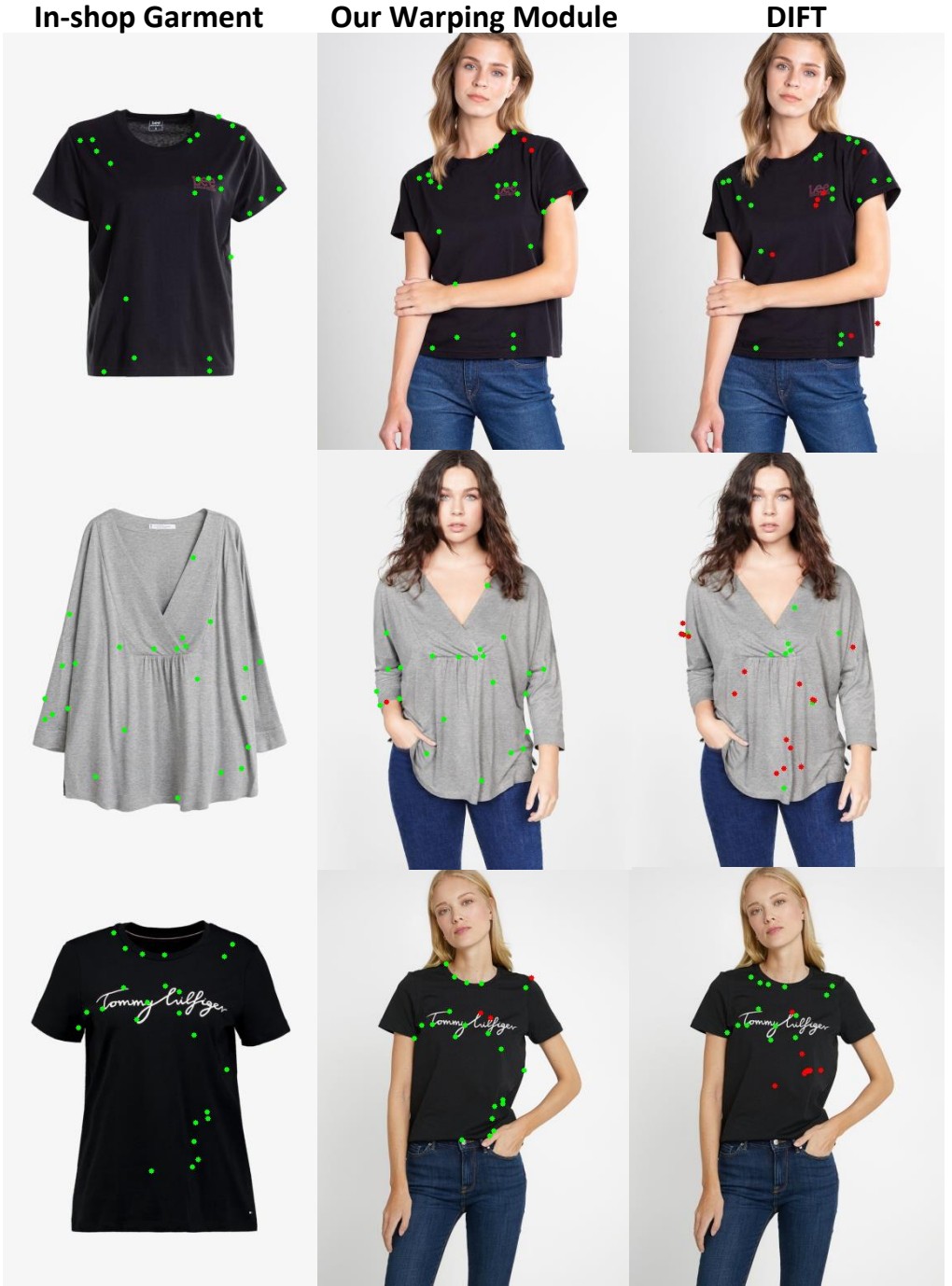

Figure 12: The visualization results of the point matching.

the visual correspondence prior in latent space is also robust to warping results (see discussion in **Discussion with warp-based methods**). This facilitates VTON of loose garments in local detail preservation. Both Figure 7 and Figure 13 demonstrate the effectiveness of our SPM-Diff for loose clothes like dresses or skirts.

## A.3 FULL-BODY VIRTUAL TRY-ON

Our SPM-Diff can naturally support full-body outfits by successively performing VTON of upper and lower garments one by one, and Fig. 14 shows some VTON results for full-body outfits.

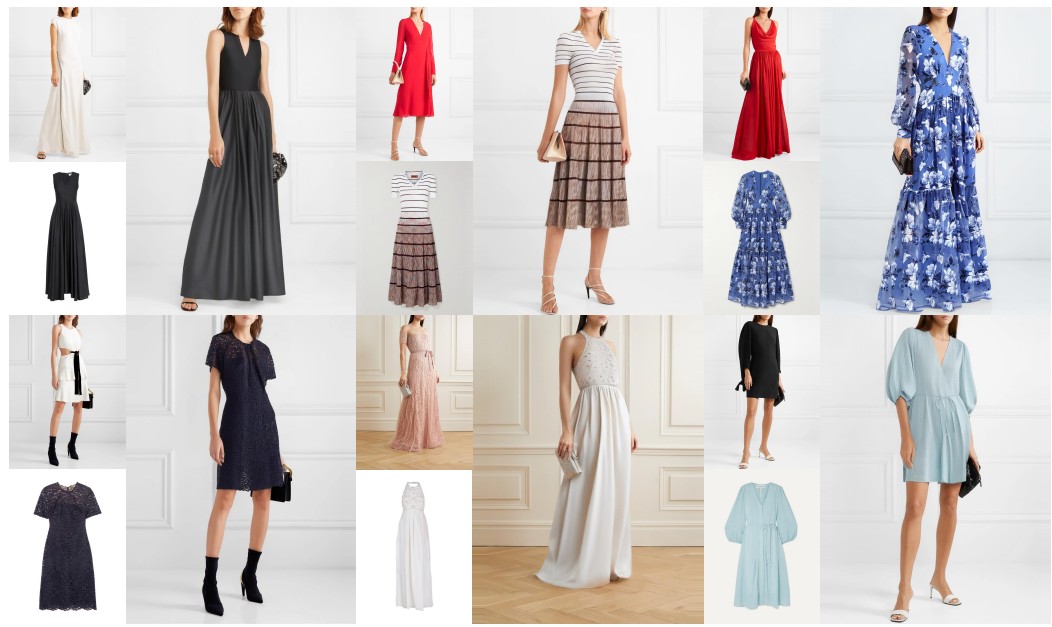

Figure 13: Tryon result for loose clothes like dresses or skirts.

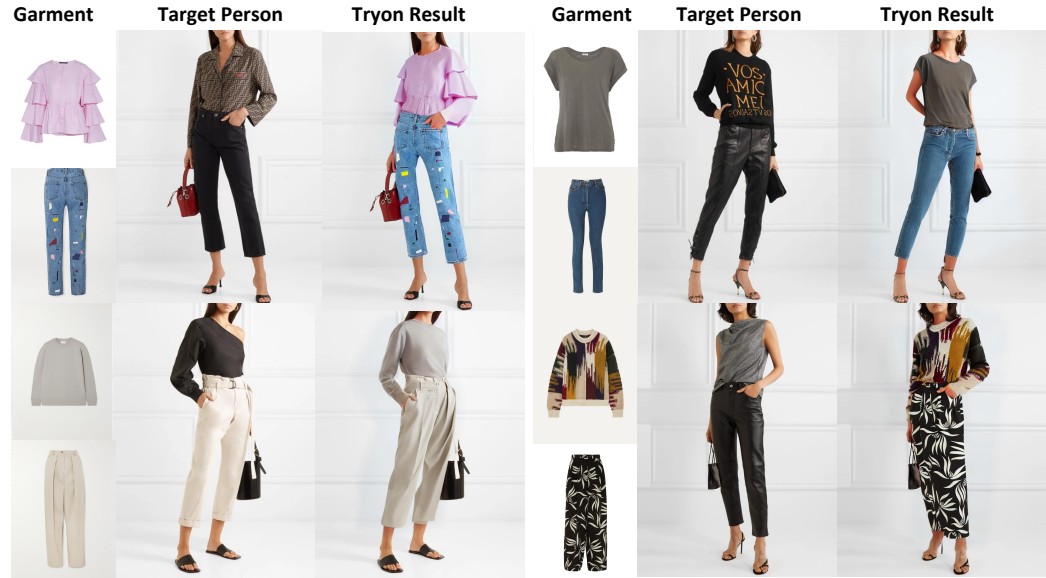

Figure 14: Tryon result of full-body outfits generated by our SPM-Diff.

### A.4    MORE DETAILS OF SPM-DIFF.

**Flow Warping Process.** As shown in Fig. 15, given in-shop garment $I_g$ and the condition triplet $C$ of target human, we leverage two feature pyramid networks, garment feature extraction $E_g(\cdot)$ and person feature extraction $E_c(\cdot)$ to extract multi-scale features, denoted as $E_g(I_g) = \{g_1, g_2, \cdots, g_N\}$ and $E_c(C) = \{c_1, c_2, \cdots, c_N\}$. These pyramid features are then fed into $N$ fusion blocks to perform local flow warping. Note that these fusion blocks exploit local-flow global-parsing blocks to estimate $N$ multi-scale local flows $f_i$, yielding the final estimated displacement field $\mathbf{M}_{G \to H}$.

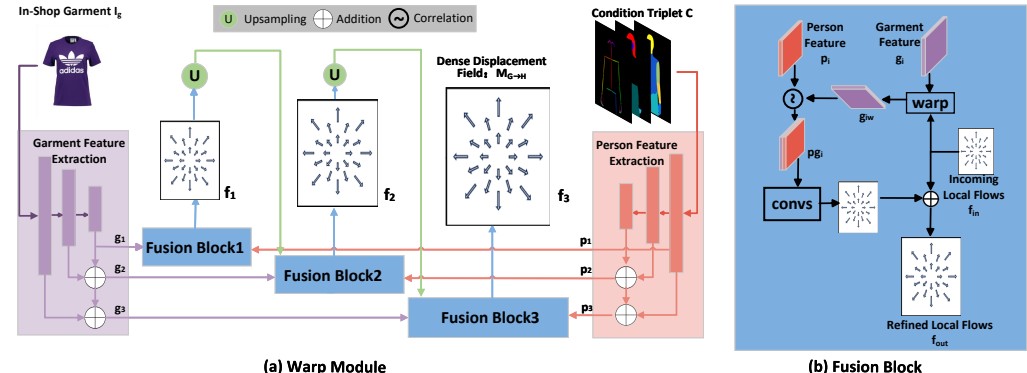

Figure 15: Overview of the flow warping module.

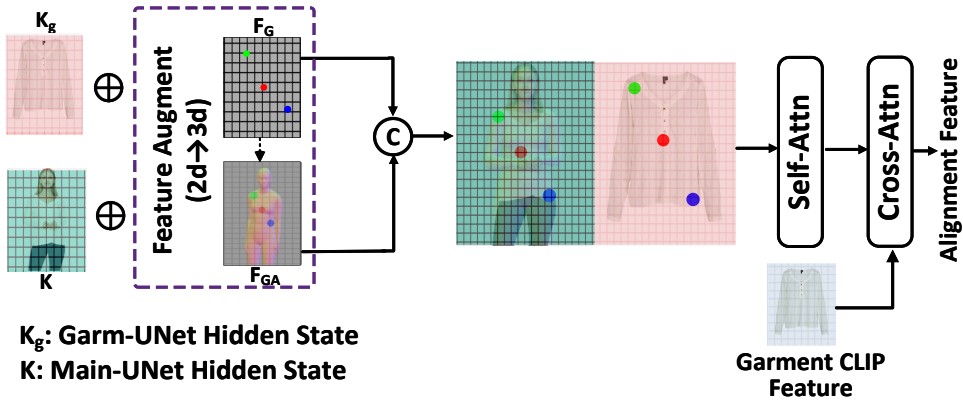

Figure 16: Details of the feature injection process.

**Feature Injection Process.** As shown in Fig. 16, the projected geometry features $\mathbf{F}_{GA}$ and garment features $\mathbf{F}_G$ of semantic points are added to the intermediate hidden states of Main-UNet (i.e., $\mathbf{K}$) and Garm-UNet (i.e., $\mathbf{K}_g$), respectively, which are further concatenated together. Finally, the derived features are further fed into the self-attention layer in Main-UNet.

## A.5 LIMITATIONS

Although our SPM-Diff nicely preserves most texture details of in-shop garments, this approach still suffers from artifacts in human hands and fails to exactly preserve other decorative items of target person, when garment overlaps with decorative items. Taking Fig. 17 as an example, the primary handbag of target person becomes larger with extra parts in the output synthesized person image. Meanwhile, some artifacts are observed on human hands. We speculate that these artifacts might be caused by the stochasticity in diffusion model, and VTON techniques only focus on the preservation of in-shop garment, while leaving the decorative items of target persons unexploited. One possible solution is to inject more conditions of human hands (e.g., hand pose) and decorative items (e.g., the canny edge information), and we leave it as one of future works.

## A.6 ETHICS STATEMENT.

We have thoroughly read the ICLR 2025 Code of Ethics and Conduct, and confirm our adherence to it. Our proposed SPM-Diff is originally devised to achieve state-of-the-art VTON performances with a novel semantic point matching and thereby enhance the online shopping experience in E-commerce industry, which poses no threat to human beings or the natural environment. However, these synthesized person images risk exacerbating misinformation and deepfake. To alleviate the misuse of our model, we will require that users adhere to usage guidelines, and meanwhile integrate

**In-shop Garment**     **Target Person**     **VTON Result**

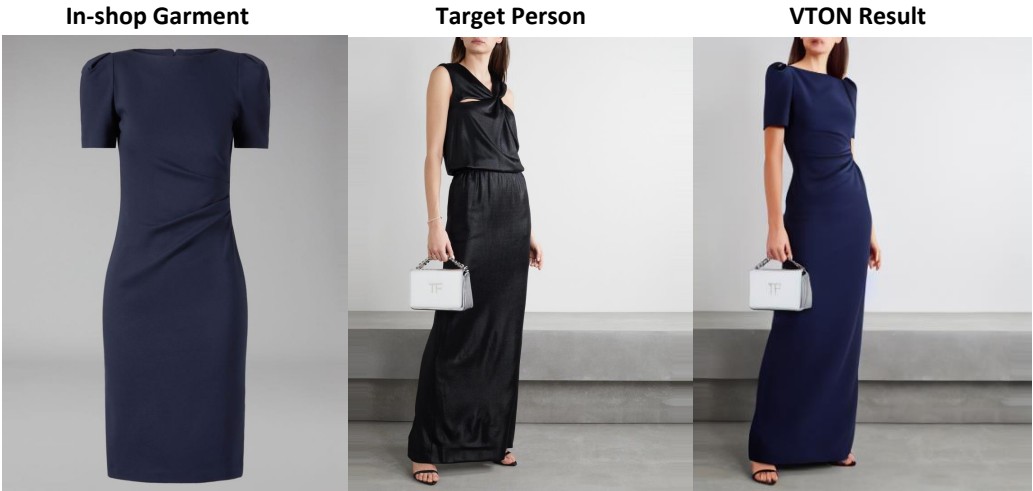

Figure 17: SPM-Diff suffers from artifacts in human hands and fails to fully preserve decorative items where they overlap with garment.

our model with an additional pipeline to automatically add digital watermark into the synthesized images. The limitations of SPM-Diff are included in Appendix A.5. All references involved in this work have been properly cited to the best of our ability. Datasets and pre-trained models are used in ways consistent with their licences in this work. The user study is conducted under appropriate ethical approvals.

