# OpenReview forum: "Incorporating Visual Correspondence into Diffusion Model for Virtual Try-On"
_ICLR.cc/2025/Conference — ICLR 2025 Poster_

### Official Review · Reviewer_THne · 2024-11-01

**Soundness:** 3
**Presentation:** 3
**Contribution:** 2
**Rating:** 6
**Confidence:** 4

**Summary:**

This paper presents a new method for virtual try-on by enforcing explicit correspondence through structured semantic points, which are first extracted from the garment image and then warped into the target body pose through local flow warping. The pair of points then serve as priors to guide the overall generation process of the try-on diffusion model, composed of a garment reference net and a main generation net. Experiments demonstrate this method generates high-quality try-on results.

**Strengths:**

1.	The proposed method is able to generate high-quality virtual try-on results. Experiments show that the proposed method outperforms existing pipelines.
2.	The idea of the paper that adopts pair of semantic points to facilitate the generation process and serve as an extra supervision signal is interesting and reasonable. If robust corresponding points could be found between the garment and the target body, they could be very good priors to boost the diffusion process.
3.	The paper is well-written and easy to follow.

**Weaknesses:**

The main weakness of the paper is on semantic point matching.
a)	How to acquire robust and accurate point matching should be the focus of the paper. However, this paper did not address this problem but rely on the local warping method proposed in GP-VTON[1], which is another virtual try-on method. This raises the concern about the contribution.
b)	The evaluation of the accuracy of the point-matching method is very limited. The point matching should achieve much more accurate results than the base diffusion model thus it could be used as a prior to guide the generation process. However, this is not demonstrated in the paper.

[1] GP-VTON: Towards General Purpose Virtual Try-on via Collaborative Local-Flow Global-Parsing Learning https://arxiv.org/abs/2303.13756

**Questions:**

There are a few questions apart from the weakness:
1.	As the local flow warping adopted in the paper relies on the coarse human body estimated with SMPL, I doubt if the warping method is capable of dealing with loose clothes like dresses or skirts. If not, will the proposed point prior still work to facilitate the generation process and get a good result?
2.	In lines 262-264, the author claims that pre-trained garment/geometry feature encoders are adopted to extract features for the following generation process, but I could not find any clarification on how these encoders were obtained. The author should clarify this.

---

> ### Author Response · Authors · 2024-11-24
> **Response to Review THne (Part 1/2)**
>
> We sincerely appreciate the kind comments from Reviewer THne. We also thank Reviewer THne for recognizing the merits in our paper (e.g., the idea is interesting and reasonable), and your kind patience in engaging in this rebuttal. Let us address some key concerns here. We hope the information here may further supplement our main submission and justify the soundness of SPM-Diff. We will include the primary results and discussion here in the main paper.
>
>
> **Q1: Contribution.**
>
> **A1:** Appreciate the comment. Existing warping-based methods (e.g., the mentioned GP-VTON) commonly adopt warping model to warp the input garment according to the input person image, and then directly leverage the warped garment image as **hard/strong condition** to generate VTON results. This way heavily relies on the accuracy of garment warping, thereby easily resulting in unsatisfactory VTON results given the inaccurate warped garments under challenging human poses (see the results of GP-VTON in [Fig. E](https://github.com/abcd1905/iclr2025-spm/blob/a3a2f0a84d1ec31d4795b0933946a3f3102f850f/figE.pdf)). On the contray, our proposed SPM-Diff adopts a soft way to exploit the visual correspondence prior learnt via the warping model as a **soft condition** to boost VTON. Technically, our SPM-Diff injects the local semantic point features of the input garment into the corresponding positions on the human body according to a flow map estimated by the warping model. This way nicely sidesteps the inputs of holistic warped garment image with amplified pixel-level projection errors, and only emphasizes the visual correspondence of the most informative semantic points between in-shop garment and output synthetic person image. Note that such visual correspondence of local semantic points are exploited in latent space (corresponding to each local region), instead of precise pixel-level location. Thus when encountering mismatched points with slight displacements within local region, SPM-Diff still leads to similar geometry/garment features as matched points, making the visual correspondence more noise-resistant (i.e., more robust to warping results). As shown in [Fig. E](https://github.com/abcd1905/iclr2025-spm/blob/a3a2f0a84d1ec31d4795b0933946a3f3102f850f/figE.pdf), given warping results with severe distortion, our SPM-Diff still manages to achieve promising results, which basically validate the effectiveness of our proposal. We will add the results and discussion in revision.
>
>
> **Q2: Evaluation of the point-matching method.**
>
> **A2:** Thanks for this point. As suggested, we examine the point matching capability of the pre-trained warping module adopted in our SPM-Diff, in comparison with a vanilla stable diffusion model. Note that here we use a training-free approach (DIFT [1]) as baseline that exploits the stable diffusion model to match the corresponding points between two images. Specifically, we construct the point matching test set by manually annotating semantic points on a subset of garment images and their corresponding points on the person image. Next, given the target semantic points on the garment image, our adopted warping module and DIFT perform point matching on the person image. Finally, we report the mean square error (MSE) between the matched points via each run and the annotated ground truth. As shown in the table below, our adopted warping module demonstrates stronger point matching capability than the vanilla stable diffusion model. In addition, we visualize the point matching results of each run in [Fig.F](https://github.com/abcd1905/iclr2025-spm/blob/a3a2f0a84d1ec31d4795b0933946a3f3102f850f/figF.pdf), and our adopted warping module yields more accurate point-matching results than vanilla stable diffusion model. We will add these results and discussions in revised version.
>
> |Model|DIFT [1]|Our Warping Model|
> |-----|-----|------|
> |$MSE\ (\downarrow)$|   39.46    |29.84|
>
> (To be continued in **Part 2/2**).

---

> ### Author Response · Authors · 2024-11-24
> **Response to Review THne (Part 2/2)**
>
> (continued from Part 1/2)
>
>
> **Q3: Semantic point matching for loose garments.**
>
> **A3:** Yes, we agree that for the cases of loose garments like dresses or skirts, some semantic points of the garment may fail to exactly fall on the human body. However, these points still convey certain local details of the garment (garment features), and the visual correspondence prior in latent space is also robust to warping results (see discussion in **Q1**). This facilitates VTON of loose garments in local detail preservation. Both Figure 7 in the appendix and the attached [Fig. G](https://github.com/abcd1905/iclr2025-spm/blob/a3a2f0a84d1ec31d4795b0933946a3f3102f850f/figG.pdf) demonstrate the effectiveness of our SPM-Diff for loose clothes like dresses or skirts.
>
>
> **Q4: Details of garment/geometry feature encoders.**
>
> **A4:** To be clear, the garment/geometry feature encoder shares similar model structure as Garment/Main-UNet. As introduced in L363-365, we jointly pre-train these two encoders in a basic dual-branch network with additional depth/normal map as inputs but without our proposed SPM mechanism, i.e., the ablated run Base+Geo map in Table 4 of the main paper. Then, the pre-trained garment/geometry feature encoder is utilized to extract garment/geometry point features, which are further injected into the self-attention layers of Main-UNet in our SPM-Diff. We will add the details in revision.
>
>
> Reference:
>
> [1] Tang, Luming and Jia, Menglin and Wang, Qianqian and Phoo, Cheng Perng and Hariharan, Bharath. Emergent correspondence from image diffusion. In  Neurips, 2023.
>
> [2] Xu, Yuhao and Gu, Tao and Chen, Weifeng and Chen, Chengcai. Ootdiffusion: Outfitting fusion based latent diffusion for controllable virtual try-on. In arXiv preprint arXiv:2403.01779, 2024.
>
> [3] Chong, Zheng and Dong, Xiao and Li, Haoxiang and Zhang, Shiyue and Zhang, Wenqing and Zhang, Xujie and Zhao, Hanqing and Liang, Xiaodan. CatVTON: Concatenation Is All You Need for Virtual Try-On with Diffusion Models. In arXiv preprint arXiv:2407.15886, 2024.
>
> [4] Xie, Zhenyu and Huang, Zaiyu and Dong, Xin and Zhao, Fuwei and Dong, Haoye and Zhang, Xijin and Zhu, Feida and Liang, Xiaodan. Gp-vton: Towards general purpose virtual try-on via collaborative local-flow global-parsing learning. In  CVPR, 2023.

---

> > ### Comment · Reviewer_THne · 2024-11-26
> >
> > My concerns are well addressed. I've upgraded my rating.

---

> > > ### Author Response · Authors · 2024-11-26
> > > **Thanks for further comments!**
> > >
> > > Dear Reviewer THne,
> > >
> > > We are glad that our response has clarified your concerns! We also feel very grateful to the many constructive comments from Reviewer THne.
> > >
> > > Thanks again for your time!

---

### Official Review · Reviewer_SQRJ · 2024-11-01

**Soundness:** 3
**Presentation:** 3
**Contribution:** 3
**Rating:** 8
**Confidence:** 5

**Summary:**

This paper presents SPM-Diff, a virtual try-on framework utilizing diffusion models that maintain garment details and shape leveraging visual correspondence. Due to warping variability, traditional VTON methods using diffusion models have difficulty preserving garment features. SPM-Diff addresses this issue by focusing on "semantic points" in garment images that capture detailed textures and shapes. Experiments on the VITON-HD and DressCode benchmarks show that SPM-Diff achieves state-of-the-art results in virtual try-on.

**Strengths:**

1. By identifying and aligning stable "semantic points" between garment and human images, SPM-Diff effectively reduces randomness in diffusion models, enabling precise and accurate garment reproduction.

2. The incorporation of 3D depth and normal maps enhances realism by accurately controlling garment fit over the body, reflecting considerable thought in modeling garment behavior in 3D space.

**Weaknesses:**

1. SPM-Diff's dependence on semantic points may lead to instability when many points are used due to interpolation and projection errors, as discussed regarding point count sensitivity.

2. SPM-Diff relies heavily on accurate depth and normal maps, which may limit its generalization to datasets or images without dependable 3D cues.

3. Although the model effectively preserves garment details, evaluations primarily concentrate on image quality metrics, neglecting user-centered metrics such as perceived realism in end-user studies.

4. Introducing "visual correspondence" into diffusion for virtual try-on is not new. Please clarify the "Semantic Point Matching" with "Warping parameters learning"

**Questions:**

The core idea of this paper is to integrate visual correspondence into the diffusion model. However, visual correspondence is not a new thing that is widely used in traditional try-ons. In addition, some people used warping + diffusion, such as : WarpDiffusion: Efficient Diffusion Model for High-Fidelity Virtual Try-on.

Please clarify their differences and provide more details about the novelty, not just replace "warping" with "visual correspondence".

---

> ### Author Response · Authors · 2024-11-24
> **Response to Review SQRJ (Part 1/2)**
>
> We sincerely appreciate the kind comments from Reviewer SQRJ. We also thank Reviewer SQRJ for recognizing the merits in our paper (e.g., precise and accurate garment reproduction), and your kind patience in engaging in this rebuttal. Let us address some key concerns here. We hope the information here may further supplement our main submission and justify the soundness of SPM-Diff. We will include the primary results and discussion here in the main paper.
>
> **Q1: Sensitivity of semantic point count.**
>
> **A1:** Thanks for this point. Yes, we agree that significantly increasing semantic point count (e.g., from $K = 25$ to $K = 50$ or $75$) will result in redundancy and interpolation/projection error, leading to degraded results as discussed in the main paper. However, when taking an in-depth look at the sensitivity of semantic point count around the optimal value $K = 25$, the semantic point count is not sensitive. To validate this, we experimented by varying semantic point count $K$ in the range of $[20, 40]$ within an interval of 5 on VITON-HD dataset. As shown in the table below, the performance under each metric only fluctuates within the range of 0.1, which basically validates that the performance is not sensitive to the change of semantic point count within this range. We will include the results and discussion in the revised version.
>
> | Point Count K | ${SSIM\uparrow}$ | ${LPIPS\downarrow}$ |${FID\downarrow}$| ${KID\downarrow}$|
> | -------- | -------- | -------- |------- |-----|
> | $K = 20$ |0.903   |  0.061 |  8.201 |0.66|
> | $K = 25$ |0.911   |  0.063 |  8.202 |0.67|
> | $K = 30$ |0.908   |  0.060 |  8.210 |0.65|
> | $K = 35$ |0.905   |  0.062 |  8.228 |0.63|
> | $K = 40$ |0.889   |  0.066 |  8.215 |0.64|
>
>
> **Q2: Accurate 3D conditions.**
>
> **A2:** It is worth noting that the 3D cues (i.e., depth/normal map) adopted in our training and inference are the estimated results by using pre-trained 3D human reconstruction model (OSX [2]), rather than manually annotated accurate 3D conditions. Considering this comment, we conducted experiments by evaluating SPM-Diff with varied estimated 3D cues achieved from different 3D human reconstruction models (e.g., PyMAF-X [1], OSX [2], 4DHumans [3]) on VITON-HD dataset. The results are summarized in the table below, and similar performances are attained across different pre-trained 3D human reconstruction models.
>
> | Model| $SSIM\uparrow$ | $LPIPS\downarrow$ |$FID\downarrow$| $KID\downarrow$|
> | -------- | -------- | -------- |------- |-----|
> | PyMAF-X [1]  | 0.901 |0.064  | 8.265|0.63
> | OSX [2] |  0.911  | 0.063 | 8.202 |0.67 |
> |4DHumans [3] |0.908|0.060|8.211|0.68
>
> Moreover, to further evaluate the generalization of our SPM-Diff under challenging scenario, we take in-the-wild person images with complex backgrounds as inputs (in contrast to the clean backgrounds in VITON-HD dataset). As shown in [Fig. C](https://github.com/abcd1905/iclr2025-spm/blob/a3a2f0a84d1ec31d4795b0933946a3f3102f850f/figC.pdf), our SPM-Diff manages to achieve promising results.
>
> (To be continued in **Part 2/2**).

---

> ### Author Response · Authors · 2024-11-24
> **Response to Review SQRJ (Part 2/2)**
>
> (continued from Part 1/2)
>
> **Q3: User-centered metrics such as perceived realism.**
>
> **A3:** Appreciate this comment. As suggested, in addition to widely adopted metric FID for image realism evaluation, we include a new user-centered metric to access image realism in user study. In this way, we have three criteria in user study: 1) preservation of the fine-grained appearance details, 2) preservation of the garment contour across diverse human body shapes, 3) the perceived image realism. We show the additional comparison results of image realism evaluation in the attached [Fig. D](https://github.com/abcd1905/iclr2025-spm/blob/a3a2f0a84d1ec31d4795b0933946a3f3102f850f/figD.pdf), which again demonstrates the superiority of our SPM-Diff against all baselines in terms of image realism. These results and discussion will be added in revised version.
>
>
> **Q4: Contribution; warp-based methods.**
>
> **A4:** We appreciate the suggested reference WarpDiffusion [5], and we are happy to discuss the differences between our SPM-Diff and these existing warp-based methods [4, 5]. In particular, existing warp-based methods [4, 5] commonly adopt warping model to warp the input garment according to the input person image, and then directly leverage the warped garment image as **hard/strong condition** to generate VTON results. This way heavily relies on the accuracy of garment warping, thereby easily resulting in unsatisfactory VTON results given the inaccurate warped garments under challenging human poses (see the results of GP-VTON in [Fig. E](https://github.com/abcd1905/iclr2025-spm/blob/a3a2f0a84d1ec31d4795b0933946a3f3102f850f/figE.pdf)). On the contrary, our proposed SPM-Diff adopts a soft way to exploit the visual correspondence prior learnt via the warping model as a **soft condition** to boost VTON. Technically, our SPM-Diff injects the local semantic point features of the input garment into the corresponding positions on the human body according to a flow map estimated by the warping model. This way nicely sidesteps the inputs of holistic warped garment image with amplified pixel-level projection errors, and only emphasizes the visual correspondence of the most informative semantic points between in-shop garment and output synthetic person image. Note that such visual correspondence of local semantic points are exploited in latent space (corresponding to each local region), instead of precise pixel-level location. Thus when encountering mismatched points with slight displacements within local region, SPM-Diff still leads to similar geometry/garment features as matched points, making the visual correspondence more noise-resistant (i.e., more robust to warping results). As shown in [Fig. E](https://github.com/abcd1905/iclr2025-spm/blob/a3a2f0a84d1ec31d4795b0933946a3f3102f850f/figE.pdf), given warping results with severe distoration, our SPM-Diff still manages to achieve promising results, which basically validate the effectiveness of our proposal. We will add the results and discussion in revision.
>
>
> References:
>
> [1] Zhang, Hongwen and Tian, Yating and Zhang, Yuxiang and Li, Mengcheng and An, Liang and Sun, Zhenan and Liu, Yebin. Pymaf-x: Towards well-aligned full-body model regression from monocular images. In IEEE Transactions on Pattern Analysis and Machine Intelligence, 2023.
>
> [2] Lin, Jing and Zeng, Ailing and Wang, Haoqian and Zhang, Lei and Li, Yu. One-stage 3d whole-body mesh recovery with component aware transformer. In CVPR, 2023.
>
> [3] Goel, Shubham and Pavlakos, Georgios and Rajasegaran, Jathushan and Kanazawa, Angjoo and Malik, Jitendra. Humans in 4D: Reconstructing and tracking humans with transformers. In ICCV, 2023.
>
> [4] Xie, Zhenyu and Huang, Zaiyu and Dong, Xin and Zhao, Fuwei and Dong, Haoye and Zhang, Xijin and Zhu, Feida and Liang, Xiaodan. Gp-vton: Towards general purpose virtual try-on via collaborative local-flow global-parsing learning. In  CVPR, 2023.
>
> [5] Li, Xiu and Kampffmeyer, Michael and Dong, Xin and Xie, Zhenyu and Zhu, Feida and Dong, Haoye and Liang, Xiaodan and others. WarpDiffusion: Efficient Diffusion Model for High-Fidelity Virtual Try-on. In arXiv preprint arXiv:2312.03667.

---

> > ### Author Response · Authors · 2024-11-29
> > **Looking forward to your feedback.**
> >
> > Dear Reviewer SQRJ,
> >
> > It seems we have not received your feedback on our response yet. As the discussion period will end soon, we were wondering if there are anything else we could address to help further clarify? We sincerely look forward to your feedback.
> >
> > Thanks again for your time!

---

> > > ### Comment · Reviewer_SQRJ · 2024-11-29
> > >
> > > I have read the rebuttal carefully. Thanks for the detailed feedback. I appreciated the response, although the "soft condition" is not new (See the MG-VTON and ClothFlow). I believe this paper is enough to be accepted, so increase the score.

---

> > > > ### Author Response · Authors · 2024-11-30
> > > > **Thanks for your feedback!**
> > > >
> > > > We sincerely appreciate your constructive review comments and kind suggestions. Many thanks for increasing the score. We are happy that our response has addressed your concerns, and we felt inspired to try out the suggested experiments and include more discussion on technical contribution. We will also discuss the mentioned "soft condition" of MG-VTON and ClothFlow in the final version. Thanks!

---

### Official Review · Reviewer_37Fc · 2024-11-03

**Soundness:** 3
**Presentation:** 3
**Contribution:** 3
**Rating:** 6
**Confidence:** 4

**Summary:**

This paper introduces a novel approach for virtual try-on tasks, addressing the challenge of preserving garment shape and fine-grained details. Specifically, the authors propose Semantic Point Matching Diffusion (SPM-Diff), a model that leverages structured semantic points as visual correspondence cues between garments and target human models. By mapping 2D semantic points to 3D-aware cues through depth and normal maps, SPM-Diff aligns garment details closely with the target human body. Experimental results on VITON-HD and DressCode datasets demonstrate the effectiveness of SPM-Diff.

**Strengths:**

- The paper leverages semantic point matching as a prior to enhance garment shape and texture preservation.

- Extensive testing on the VITON-HD and DressCode datasets demonstrates the model's robustness and superior performance.

- The authors have provided the code to ensure reproducibility of their results.

**Weaknesses:**

- In line 254, local flow warping is used as a method to associate semantic points with their counterparts on the target person. It would be better to provide more detail on the local flow warping process for better understanding.

- For Figure 5, it’s difficult to assess the accuracy of the matched points. Using different colors (e.g. red and green) to illustrate correct and incorrect mappings would improve clarity.

- Adding a figure to illustrate feature injection would help clarify how point features are incorporated into the Main-UNet.

**Questions:**

- Could you provide more details on the local flow warping process?

- In Figure 5, are all the points shown the correct correspondences? If not, would it be possible to use different colors to distinguish correct from incorrect point mappings? Additionally, how do you assess the accuracy or correctness of these point mappings?

- It would be better to include a figure to illustrate the feature injection process, specifically showing how point features are integrated into the Main-UNet?

---

> ### Author Response · Authors · 2024-11-24
> **Response to Reviewer 37Fc**
>
> We sincerely appreciate the kind comments from Reviewer 37Fc. We also thank Reviewer 37Fc for recognizing the merits in our paper (e.g., the model's robustness and superior performance), and your kind patience in engaging in this rebuttal. Let us address some key concerns here. We hope the information here may further supplement our main submission and justify the soundness of SPM-Diff. We will include the primary results and discussion here in the main paper.
>
>
> **Q1: Details of local flow warping process.**
>
> **A1:** To be clear, the local flow warping module is employed to estimate the dense displacement field $M_{G \rightarrow H} \in \mathbb{R}^{H \times W \times 2}$ from in-shop garment $I_g$ to the target human (i.e., the condition triplet $C$ comprising human pose, densepose pose, and preserved region parsing). Each element $M_{G \rightarrow H}^{(i,j)}$ represents the relative displacement (i.e., 2D offset vector) for each point at coordinate $(i, j)$ on in-shop garment $I_g$ relative to the person image $I_p$. Here we employ flow warping module like GP-VTON to estimate the dense displacement field in between. Formally, given in-shop garment $I_g$ and the condition triplet $C$ of target human, we leverage two feature pyramid networks, garment feature extraction $E_g(\cdot)$ and person feature extraction $E_c(\cdot)$ to extract multi-scale features, denoted as $E_g(I_g) = \{g_1, g_2, \cdots , g_N\}$ and $E_c(C) = \{c_1, c_2, \cdots, c_N\}$. These pyramid features are then fed into $N$ fusion blocks to perform local flow warping. Note that these fusion blocks exploit local-flow global-parsing blocks to estimate $N$ multi-scale local flows $f_i$, yielding the final estimated displacement field $M_{G \rightarrow H}$. Considering this comment, we will include an additional figure to depict the technical details of local flow warping process ([Fig. A](https://github.com/abcd1905/iclr2025-spm/blob/a3a2f0a84d1ec31d4795b0933946a3f3102f850f/figA.pdf)) in the revised version.
>
>
> **Q2: Matched points in Figure 5.**
>
> **A2:** To clarify, in Figure 5, all green point pairs/mappings basically reflect the correct correspondences between the generated image and the input garment. Note that here we follow the typical point mapping method in SuperPoint[1] to perform HPatches homography estimation (i.e., the estimation of correst matches). Specifically, we perform nearest neighbor matching between all interest points and their descriptors detected in the in-shop garment image and those in the synthesized person image. Then we employ OpenCV's implementation of findHomography() with RANSAC to compute the correctly matched points in the image pairs, which are further marked in green. In this way, the denser the green point mappings covered over two images, the better the perservation of interest points (i.e., garment details) for synthesized person image.
>
>
> **Q3: Add figure of feature injecting process.**
>
> **A3:** Appreciate this comment. As suggested, we include an additional [Fig. B](https://github.com/abcd1905/iclr2025-spm/blob/a3a2f0a84d1ec31d4795b0933946a3f3102f850f/figB.pdf) to illustrate the feature injection process. Technically, given the geometry features $F_{GA}^l$ and garment features $F_{G}^l$ of semantic points, we first augment intermediate hidden states of Garm-UNet/Main-UNet (i.e., $K^l$/$K_g^l$) with $F_{GA}^l$/$F_{G}^l$. Then the augmented intermediate features of Garm-UNet/Main-UNet are concatenated, followed with $l$-th self-attention layer in Main-UNet:
> $$
> Attn(Q^l,K^l_{*},V^l,F_{GA}^l,F_{G}^l) = Softmax(\frac{(Q^l + F_{GA}^l) \cdot [K^l + F_{GA}^l, K^l_g + F_{G}^l]^T}{\sqrt{d}}V^l).
> $$
> In this way, the hidden states of Main-UNet are strengthened with additional geometry \& garment features of semantic points and intermediate features of Garm-UNet through feature injection.
>
> Reference: [1] DeTone, Daniel and Malisiewicz, Tomasz and Rabinovich, Andrew. Superpoint: Self-supervised interest point detection and description. In CVPR Workshop, 2018.

---

> > ### Comment · Reviewer_37Fc · 2024-11-26
> >
> > Thanks for the author's reply. After reading the rebuttal and other reviewers' comments, I also agree that the semantic point mapping part is not novel enough. Thus I leave my rating unchanged.

---

> > > ### Author Response · Authors · 2024-11-26
> > > **Thanks for your response**
> > >
> > > Appreciate your response. As discussed in **Q1** of Reviewer THne (we are glad to see our response has clarified the concern of Reviewer THne who has just upgraded the rating), we would like to emphasize that our SPM-Diff is compeletely different from existing warping-based methods:
> > >
> > > Existing warping-based methods (e.g., GP-VTON) commonly adopt warping model to warp the input garment according to the input person image, and then directly leverage the warped garment image as **hard/strong condition** to generate VTON results. This way heavily relies on the accuracy of garment warping, thereby easily resulting in unsatisfactory VTON results given the inaccurate warped garments under challenging human poses (see the results of GP-VTON in [Fig. E](https://github.com/abcd1905/iclr2025-spm/blob/a3a2f0a84d1ec31d4795b0933946a3f3102f850f/figE.pdf)). On the contray, our proposed SPM-Diff adopts a soft way to exploit the visual correspondence prior learnt via the warping model as a **soft condition** to boost VTON.
> > >
> > > Technically, although warping model is adopted, we novelly inject the local semantic point features of the input garment into the corresponding positions on the human body according to a flow map estimated by the warping model. This way nicely sidesteps the inputs of **holistic warped garment image** with amplified pixel-level projection errors, and only emphasizes the visual correspondence of **the most informative semantic points** between in-shop garment and output synthetic person image. Note that such visual correspondence of local semantic points are exploited in **latent space** (corresponding to each local region), instead of precise **pixel-level location**. Thus when encountering mismatched points with slight displacements within local region, SPM-Diff still leads to similar geometry/garment features as matched points, making the visual correspondence more noise-resistant (i.e., more robust to warping results). As shown in [Fig. E](https://github.com/abcd1905/iclr2025-spm/blob/a3a2f0a84d1ec31d4795b0933946a3f3102f850f/figE.pdf), given warping results with severe distortion, our SPM-Diff still manages to achieve promising results, which basically validate the effectiveness of our proposal. We have added these results and discussion in revision.

---

> ### Author Response · Authors · 2024-11-29
> **We look forward to your feedback**
>
> Dear Reviewer 37Fc,
>
> It seems we have not received your further feedback on our updated response on technical contribution. Thus here we provide a more detailed clarification on technical contribution of our proposed semantic point mapping, especially compared to existing warping-based methods. Though warping module is exploited to estimate the dense displacement between garment and target person, the success of our work is more than that. Our contributions are from three aspects:
>
> i. The input condition of diffusion model is new. Unlike existing warping-based models directly leverage **holistic warped garment image** as inputs to trigger diffusion process, our work seeks **the most informative semantic points** as a new condition to enable VTON task. Such semantic point inputs go beyond traditional holistic warped inputs by emphasizing unique regional fine-grained texture details, thereby encouraging better preservation of fine-grained details in VTON.
>
> ii. The injection of additional condition into diffusion model is novel. Existing warping-based methods commonly take the warped garment image as **hard/strong condition** to guide VTON. This way heavily relies on the accuracy of garment warping, thereby easily resulting in unsatisfactory VTON results given the inaccurate warped garments under challenging human poses (see the results of GP-VTON in [Fig. E](https://github.com/abcd1905/iclr2025-spm/blob/a3a2f0a84d1ec31d4795b0933946a3f3102f850f/figE.pdf)). On the contray, our proposed SPM-Diff adopts a soft way to exploit the visual correspondence prior derived from semantic points as a **soft condition** to boost VTON. Note that such visual correspondence of local semantic points are exploited in **latent space** (corresponding to each local region), instead of precise alignment of **pixel-level location** in typical warping-based models. Thus when encountering mismatched points with slight displacements within local region, SPM-Diff still leads to similar geometry/garment features as matched points, making the visual correspondence more noise-resistant (i.e., more robust to warping results). As shown in [Fig. E](https://github.com/abcd1905/iclr2025-spm/blob/a3a2f0a84d1ec31d4795b0933946a3f3102f850f/figE.pdf), given warping results with severe distortion, our SPM-Diff still manages to achieve promising results.
>
> iii. To amplify our proposed semantic point matching along the whole diffusion process, we devise a point-focused diffusion objective that emphasizes on the reconstruction of semantic points over target persons (pursuing better preservation of fine-grained texture details), rather than typical reconstruction of holistic target person image.
>
> We therefore kindly invite Reviewer 37Fc to reconsider assessment on our SPM-Diff's essential technical contributions, depending on the above discussions.

---

> > ### Comment · Reviewer_37Fc · 2024-11-30
> >
> > Thank you for providing additional details on your contributions. I have increased my rating accordingly. I look forward to seeing the potential integration of this method into real-world products.

---

> > > ### Author Response · Authors · 2024-11-30
> > > **Thanks for your response！**
> > >
> > > We sincerely appreciate your valuable time on our response! We are very glad to see our answers have further clarified the concerns. Thanks again for your constructive comments and suggestions!

---

### Author Response · Authors · 2024-11-24
**Revision uploaded**

Dear Reviewers,

We sincerely appreciate the constructive comments from everyone, and we enjoyed this process pretty much! In order to reflect the required empirical evidence in the pdf submission, we have uploaded a pdf revision (blue colored highlights), by following the ICLR 2025 rules. We understand that we are allowed to submit pdf revisions by November 27, so please kindly let us know if any additional result is required in the pdf by this date.

Best Regards

---

### Meta-Review · Area_Chair_QMWV · 2024-12-17

**Metareview:**

This paper receives unanimous positive ratings of 6,8,6. The AC follows the recommendations of the reviewers to accept the paper. The reviewers think that the proposed method is effective in giving high quality results. Extensive experimental results on various datasets are also shown to support the model's robustness and effectiveness. The authors have provided the code to ensure reproducibility of their results. Although a reviewer was initially concerned about semantic point matching, the authors managed to clarify the doubts in the rebuttal and discussion phases. The reviewer finally raised the score to positive.

**Additional Comments On Reviewer Discussion:**

Although a reviewer was initially concerned about semantic point matching, the authors managed to clarify the doubts in the rebuttal and discussion phases. The reviewer finally raised the score to positive.

---

### Decision · Program_Chairs · 2025-01-22

Accept (Poster)